# Pyrin dephosphorylation is sufficient to trigger inflammasome activation in familial Mediterranean fever patients

Flora Magnotti[1], Lucie Lefeuvre[1,2], Sarah Benezech[1,2,‡] (ID), Tiphaine Malsot[1], Louis Waeckel[1,2], Amandine Martin[1], Sébastien Kerever[3], Daria Chirita[1], Marine Desjonqueres[2,4], Agnès Duquesne[2,4], Mathieu Gerfaud-Valentin[2,5], Audrey Laurent[2,4], Pascal Sève[2,5], Michel-Robert Popoff[6], Thierry Walzer[1], Alexandre Belot[1,2,4], Yvan Jamilloux[1,2,5,*,†] (ID) & Thomas Henry[1,**,†] (ID)

## Abstract

**Familial Mediterranean fever (FMF) is the most frequent hereditary systemic autoinflammatory syndrome. FMF is usually caused by biallelic mutations in the *MEFV* gene, encoding Pyrin. Conclusive genetic evidence lacks for about 30% of patients diagnosed with clinical FMF. Pyrin is an inflammasome sensor maintained inactive by two kinases (PKN1/2). The consequences of *MEFV* mutations on inflammasome activation are still poorly understood. Here, we demonstrate that PKC superfamily inhibitors trigger inflammasome activation in monocytes from FMF patients while they trigger a delayed apoptosis in monocytes from healthy donors. The expression of the pathogenic p.M694V *MEFV* allele is necessary and sufficient for PKC inhibitors (or mutations precluding Pyrin phosphorylation) to trigger caspase-1- and gasdermin D-mediated pyroptosis. In line with colchicine efficacy in patients, colchicine fully blocks this response in FMF patients' monocytes. These results indicate that Pyrin inflammasome activation is solely controlled by Pyrin (de)phosphorylation in FMF patients while a second control mechanism restricts its activation in healthy donors/non-FMF patients. This study paves the way toward a functional characterization of *MEFV* variants and a functional test to diagnose FMF.**

**Keywords** autoinflammation; caspase-1; colchicine; diagnosis; pyroptosis
**Subject Categories** Genetics, Gene Therapy & Genetic Disease; Immunology

## Introduction

Familial Mediterranean fever (FMF) is the most frequent hereditary systemic autoinflammatory disorder characterized by recurrent episodes of fever, serositis, and abdominal pain (Sonmez *et al*, 2016). Its feared complication is secondary AA amyloidosis, which can lead to end-stage kidney disease. Colchicine, an inhibitor of microtubule polymerization, decreases chronic inflammation and represents the cornerstone of FMF treatment (Goldfinger, 1972). Daily and lifelong administration of colchicine is currently recommended for FMF patients.

Familial Mediterranean fever diagnosis relies first on clinical criteria. Due to the absence of pathognomonic clinical signs and to heterogeneity in clinical presentations (Mor *et al*, 2003; Padeh *et al*, 2010), FMF diagnosis can be challenging (Giancane *et al*, 2015). Familial Mediterranean fever is associated with mutations in the *MEFV* gene. Mendelian transmission of the disease occurs mostly in an autosomal recessive mode. As of today, genetic screening confirms the FMF diagnosis upon identification of biallelic mutations in clearly pathogenic *MEFV* variants (Shinar *et al*, 2012). Nine sequence variants of *MEFV* are considered clearly pathogenic (Shinar *et al*, 2012). Yet, there are 365 *MEFV* variants listed in the Infevers database (Sarrauste de Menthiere *et al*, 2003), most of them of uncertain significance, which can result in misdiagnosis or diagnosis delay (Lidar *et al*, 2005). Furthermore, a substantial proportion of clinically diagnosed FMF patients (up to 30%) presents only a single *MEFV* pathogenic variant (Dode *et al*, 2000; Lachmann *et al*, 2006; Jeru *et al*, 2013). Finally, no *MEFV* variant is found in 5–14% of clinically diagnosed

1  CIRI, Centre International de Recherche en Infectiologie, Inserm, U1111, Université Claude Bernard Lyon 1, CNRS, UMR5308, École Normale Supérieure de Lyon, Univ. Lyon, Lyon, France
2  Hospices Civils de Lyon, Lyon, France
3  Department of Anesthesiology and Critical Care, St Louis-Lariboisière University Hospital, AP-HP, ECSTRA Team, Epidemiology and Biostatistics, Sorbonne Paris Cité Research Centre, UMR 1153, Inserm, University Denis Diderot-Paris VII, Paris, France
4  Service de Néphrologie, Rhumatologie, Dermatologie pédiatriques, HFME, Bron, France
5  Service de Médecine Interne, Hôpital de la Croix-Rousse, Lyon, France
6  Bacterial Toxins, Institut Pasteur, Paris, France
   *Corresponding author. Tel: +33 4 37 28 23 72; E-mail: yvan.jamilloux@inserm.fr
   **Corresponding author. Tel: +33 4 37 28 23 72; E-mail: thomas.henry@inserm.fr
   †These authors contributed equally to this work as senior authors
   ‡Present address: Institut d'Hématologie et Oncologie Pédiatrique, Lyon, France

FMF patients (Lachmann et al, 2006; Toplak et al, 2012). Due to all these situations, genetic testing has a 70–80% positive predictive value (Soriano & Manna, 2012) and the median delay between disease onset and diagnosis remains long (1.4 years for patients born in the 21st century; Toplak et al, 2012). Furthermore, the generalization of next-generation sequencing leads to the identification of novel rare variants of unknown impact. Functional assays robustly discriminating pathogenic MEFV variants from non-pathogenic MEFV polymorphisms are needed to sustain diagnosis and the development of personalized medicine (Van Gorp et al, 2016).

MEFV encodes Pyrin, an inflammasome sensor detecting Rho A GTPase inhibition (Xu et al, 2014). Inactivation of Rho A by various bacterial toxins triggers activation of the Pyrin inflammasome, i.e., oligomerization of the inflammasome adaptor ASC, caspase-1 activation, secretion of the pro-inflammatory cytokines IL-1β and IL-18, and an inflammatory cell death termed pyroptosis (Cookson & Brennan, 2001; Martinon et al, 2002; Xu et al, 2014). At steady state, Pyrin is maintained inactive by phosphorylation of its serine residues S208 and S242. Two kinases (PKN1/2) from the PKC superfamily phosphorylate Pyrin, leading to its sequestration by 14-3-3 chaperone proteins (Gao et al, 2016; Masters et al, 2016; Park et al, 2016; Van Gorp et al, 2016). Rho A inhibition leads to dephosphorylation of Pyrin, its release from the 14-3-3 proteins and the assembly/activation of the Pyrin inflammasome. Of note, in healthy individuals, the transition from 14-3-3-free Pyrin to ASC oligomerization and Pyrin inflammasome activation requires microtubule dynamics (Gao et al, 2016). Colchicine specifically blocks the Pyrin inflammasome downstream of Pyrin release from the 14-3-3 proteins and upstream of ASC oligomerization (Gao et al, 2016). In FMF patients, the microtubule-dependent mechanism might be deficient since a recent report indicated that colchicine is inefficient to block Pyrin inflammasome activation in PBMCs from FMF patients (Van Gorp et al, 2016). This in vitro result, at odds with the clinical efficacy of colchicine in FMF patients, is still poorly understood. A two-step activation model is emerging with (i) dephosphorylation of Pyrin following inhibition of PKN1/2 and (ii) Pyrin inflammasome maturation involving a colchicine-targetable microtubule dynamics event (Gao et al, 2016). The link between the two steps remains unclear. Particularly, it is unknown whether dephosphorylation of Pyrin automatically leads to Pyrin inflammasome activation in cells with intact microtubule dynamics. Finally, the impact of MEFV mutations on each step is controversial (Gao et al, 2016; Masters et al, 2016; Park et al, 2016; Van Gorp et al, 2016).

In this work, we demonstrate that PKC superfamily inhibitors trigger inflammasome activation, IL-1β secretion, and pyroptosis in monocytes from FMF patients while they fail to do so in monocytes from healthy donors (HD) in which they trigger a delayed apoptosis. PKC superfamily inhibitor-mediated inflammasome activation was blocked by colchicine in FMF patients' monocytes in line with the efficacy of this drug in patients. The mechanism of the differential control of the Pyrin inflammasome was pinpointed to specific MEFV mutations in human monocyte cell lines expressing either one of three common clearly pathogenic MEFV variants, p.M694V, p.M694I, or p.M680I. Importantly, the cytotoxic effect of PKC superfamily inhibitors on the p.M694V allele-expressing cells could be recapitulated genetically by mutating the Pyrin Serine 242 or S208 residues. These results suggest that, while Pyrin inflammasome is controlled by two independent mechanisms in healthy donors, in FMF patients, the Pyrin inflammasome lacks one safeguard mechanism and is only regulated by Pyrin phosphorylation. Finally, our results indicate that these differences could be exploited to develop a functional diagnostic test.

# Results

## PKC inhibitors trigger IL-1β release in monocytes from FMF patients

The current model for Pyrin inflammasome activation indicates that activation results from the dephosphorylation of Pyrin following the lack of sustained activation of PKN1/2, two kinases from the PKC superfamily (Park et al, 2016). To explore the mechanisms underlying deregulation of the Pyrin inflammasome in FMF patients, we decided to assess the efficacy of staurosporine (a potent PKC superfamily inhibitor targeting PKN1/2; Davis et al, 2011) to trigger IL-1β release in primary monocytes from HD or FMF patients. We observed no to very low IL-1β release from monocytes isolated from HD in response to LPS + staurosporine (Fig 1A). In our experimental conditions, monocytes from 31 out of the 33 HD (94%) released < 50 pg/ml of IL-1β (Fig 1A). In sharp contrast, monocytes from FMF patients released moderate to high levels of IL-1β, leading to an average level 17-fold higher (422 pg/ml, P < 0.0001) than the average level in the supernatant of HD monocytes (25 pg/ml) (Fig 1A and Appendix Fig S1A for a detailed version including patients' genotype). These differences were conserved over several staurosporine concentrations and at several times post-treatment (Appendix Fig S2A and B). This result indicates strongly differing inflammasome responses to PKC superfamily inhibition between FMF patients and HD. To confirm this result, we used UCN-01, a hydroxylated derivative of staurosporine, which displays a better selectivity for PKC superfamily kinases (Tamaoki, 1991). Similar findings were observed (Fig 1B and Appendix Fig S1B) with monocytes from FMF patients releasing > 10-fold higher IL-1β levels than HD monocytes did. The same trend (Appendix Fig S2C and D) was observed using the bisindolylmaleimide RO 31–8220, another PKC superfamily inhibitor of different chemical structure (Davis et al, 1992). IL-1β levels following treatment with UCN-01 and staurosporine were significantly correlated in the different patients (Appendix Fig S2E). As seen with staurosporine, the difference in IL-1β response between monocytes from HD and FMF patients was conserved over a large range of concentrations of UCN-01 (Appendix Fig S2F). The hyper-responsiveness of FMF monocytes to PKC superfamily inhibitors thus differs from their hyper-responsiveness to Clostridioides difficile toxin TcdB, which was observed only at low doses of TcdB (Jamilloux et al, 2018).

IL-1β levels were substantially decreased upon addition, 30 min before UCN-01, of the caspase-1 inhibitors VX-765 or YVAD-FMK (Appendix Fig S2G). Neither the caspase-3 inhibitor (DEVD-FMK) nor the caspase-8 inhibitor (IETD-FMK) demonstrated a robust inhibition of IL-1β release. This result suggests that UCN-01 triggers inflammasome activation in FMF patient monocytes. As previously described (Van Gorp et al, 2016), we did not observe any difference in IL-1β release in response to engagement of the NLRP3 inflammasome by LPS + ATP (Appendix Fig S2H) or of the NLRC4 inflammasome (Jamilloux et al, 2018). Furthermore, LPS + staurosporine treatment did not lead to differential TNF secretion between monocytes from HD and FMF patients (Fig 1C), indicating that the

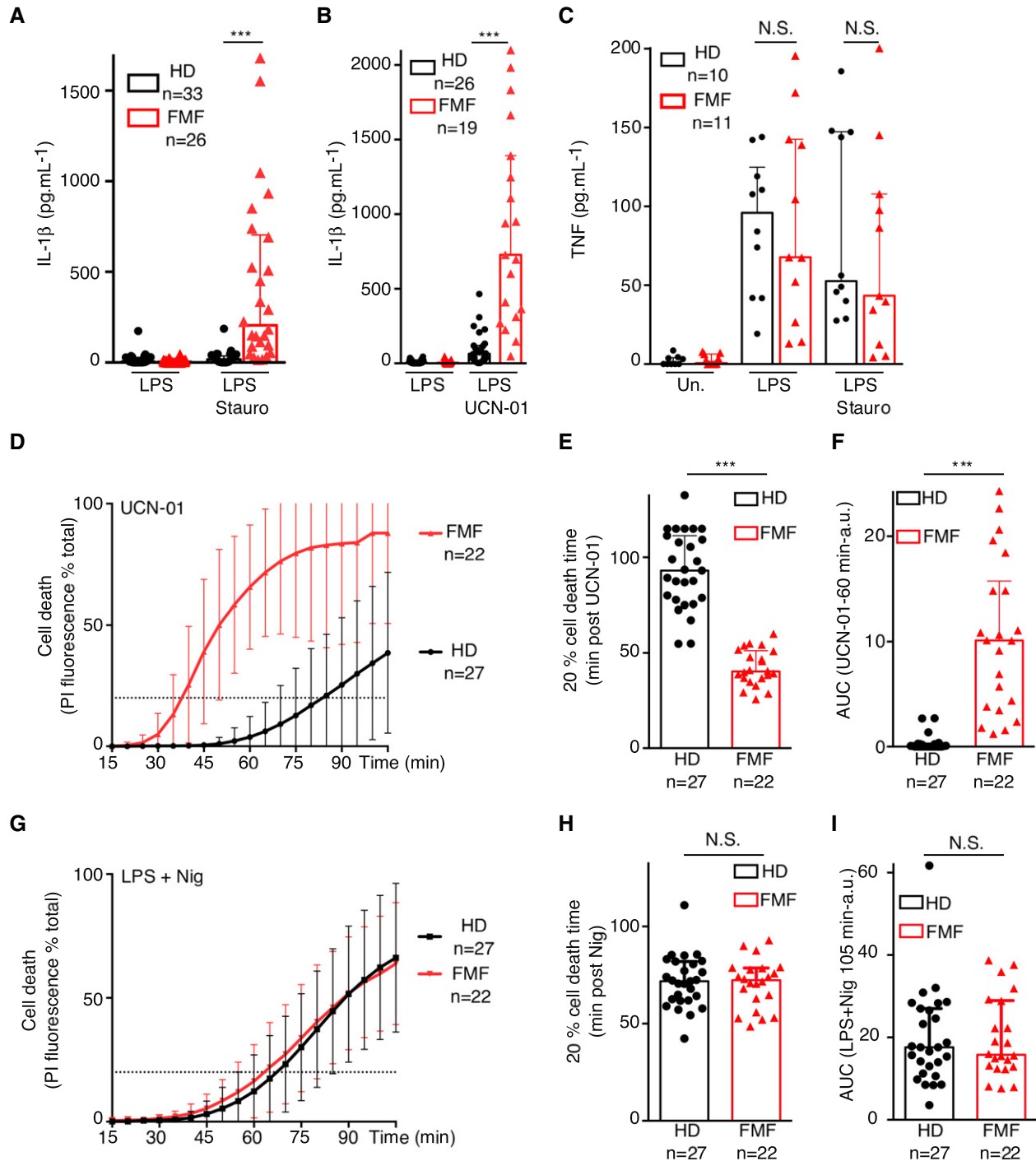

**Figure 1. PKC inhibitors specifically trigger IL-1β release and a fast cell death in monocytes from FMF patients.**

A–I  Monocytes from healthy donors (HD) or FMF patients were either primed with LPS (A–C, G–I) or not (D–F) and treated with (A, C) 1.25 μM staurosporine (Stauro), (B, D–F) 12.5 μM UCN-01, or (G–I) 5 μM nigericin (Nig). (A, B) IL-1β and (C) TNF level were quantified by ELISA at 90 min post stimulation. (D, G) Cell death was monitored in real time by measuring propidium iodide influx/fluorescence every 5 min. (E, H) The time required to reach 20% cell death and (F, I) the area under the curve (AUC) were computed for each HD or FMF patients.

Data information: (A–C, E–F, H, I) Each dot represents the mean value from three biological replicates for one HD or patient. The bar represents the median ± interquartile range. a.u.: arbitrary units. (D, G) Each point of the curve corresponds to the average cell death values from the indicated number of HD or FMF patients (for each individual, the value is the mean of a biological triplicate). The dotted line indicates the 20% cell death value. (A, B, F) ***P < 0.0001 by Wilcoxon rank-sum test. (C) One-way ANOVA with Sidak's multiple comparison tests was performed. LPS: N. S. Not significant P = 0.99; LPS + Staurosporine N.S.: P = 0.77. (E, H) Unpaired t-tests were performed, and two-tailed P-values are shown. (E) ***P < 0.0001, (H) N.S.: P = 0.79. (I) N.S. P = 0.9 by Wilcoxon rank-sum test.
Source data are available online for this figure.

differing response to PKC inhibitors between HD and FMF patients is specific to inflammasome activation. These results suggest that dephosphorylation of Pyrin is sufficient to trigger inflammasome activation in monocytes from FMF patients while a PKC-independent mechanism limits IL-1β release in HD monocytes.

## PKC inhibitors trigger fast cell death in monocytes from FMF patients

Inflammasome activation is often associated with a fast cell death process termed pyroptosis. We thus investigated whether PKC inhibitors, in the absence of LPS priming, trigger cell death in monocytes from HD and FMF patients. Indeed, UCN-01 triggered a very rapid influx of propidium iodide in monocytes from FMF patients while it was much delayed in monocytes from HD (Fig 1D). These kinetics were determined to be significantly different by quantifying the time post-UCN-01 addition leading to 20% cell death (dotted line in Fig 1D and E, $P < 0.0001$) and the area under the curve (AUC, Fig 1F, $P < 0.0001$). The difference in cell death was specific to PKC inhibitors since NLRP3 inflammasome activation by LPS + nigericin (Mariathasan *et al*, 2006) triggered propidium iodide influx with similar kinetics in monocytes from HD and FMF patients (Fig 1G–I and Appendix Fig S2I). Importantly, the UCN-01-mediated fast cell death was observed in the absence of LPS treatment, indicating that the Pyrin inflammasome does not require TLR-mediated priming, as previously demonstrated following *C. difficile* toxin treatment (Van Gorp *et al*, 2016; Jamilloux *et al*, 2018).

## PKC inhibitors differentially trigger pyroptosis or apoptosis in monocytes from FMF patients and HD

The kinetics of monocytes death and its association with IL-1β release suggest that PKC inhibitors trigger pyroptosis in monocytes from FMF patients. To strengthen this finding, we directly evaluated the ability of the inflammasome adaptor ASC to form specks as a readout of inflammasome complex formation. At 40 min post-UCN-01 treatment, monocytes were fixed and immuno-stained for ASC. More than 35% of monocytes from FMF patients displayed ASC

specks while UCN-01 treatment did not substantially increase the frequency of speck-containing cells in HD monocytes (Fig 2A and B and control experiment in Appendix Fig S3). This result indicates that the Pyrin inflammasome activation is controlled by a phosphorylation-independent mechanism upstream of ASC oligomerization in monocytes from HD and that this control mechanism is defective in FMF patients.

The UCN-01-mediated induction of inflammasome activation in monocytes from FMF patients was further confirmed by quantifying cells containing active caspase-1, using the fluorescent inhibitor probe FAM-YVAD-FMK, referred to as FLICA-Casp1. As quantified by flow cytometry, 33% of monocytes from FMF patients stained positive for FLICA-Casp1 at 40 min post-UCN-01 addition, while only 6% of monocytes from HD did, a proportion similar to the one observed in the untreated samples (Fig 2C).

These results establish that PKC inhibitors specifically trigger pyroptosis in monocytes from FMF patients. Yet, PKC inhibitors also lead to a late cell death in HD monocytes (Fig 1D). Based on the well-known activity of PKC inhibitors to trigger apoptosis (Nie *et al*, 2014), and on the absence of signs of inflammasome activation (Figs 1B and 2A–C), we hypothesized that UCN-01 might trigger apoptosis in HD monocytes. We thus monitored phosphatidyl serine externalization in propidium iodide-negative cells as an early marker of apoptosis following UCN-01 (Fig 2D and E) or staurosporine (Appendix Fig S4) treatment. At 90 min post-treatment with PKC superfamily inhibitors, a large number of HD monocytes stained positive for Annexin-V and were negative for propidium iodide, strongly suggesting that HD monocytes died by apoptosis. In contrast, we did not detect any Annexin-V-positive/propidium iodide (PI)-negative cells induced by UCN-01 in monocytes from FMF patients, which is consistent with FMF patients' monocytes dying by pyroptosis upon PKC inhibitor exposure. As expected, in response to LPS + nigericin, we did not detect a substantial proportion of Annexin-V$^+$/PI$^-$ cells, neither in monocytes from HD nor from FMF patients (Fig 2E).

Altogether, these results indicate that PKC inhibitors trigger inflammasome activation, IL-1β release, and pyroptosis in monocytes from FMF patients, while inflammasome is not activated in monocytes from HD.

**Figure 2. The PKC inhibitor, UCN-01, triggers pyroptosis or apoptosis in monocytes from FMF patients and HD, respectively.**

A, B   Monocytes from HD or FMF patients were treated with 12.5 μM UCN-01 for 40 min or primed with LPS (3 h) and treated with 5 μM nigericin (Nig) for 90 min. (A) Cells were immuno-stained for ASC. ASC specks are indicated by red arrowheads. Representative confocal microscopy images from one HD (top panels) and one FMF patient (bottom panels) are shown. Scale bars: 10 μm and 2.5 μm in the main figures and onsets, respectively. (B) Quantification of ASC specks in HD and FMF patients' monocytes by immunofluorescence.

C   The frequency of cells positive for active caspase-1 was quantified by flow cytometry, using FLICA-Caspase-1 in HD and FMF patients.

D, E   Cell death was assessed at 90 min post-UCN-01 or post-LPS + nigericin treatment by determining the percentage of Annexin-V$^+$/PI$^-$ cells and of PI$^+$ cells among dead cells (Annexin-V$^+$ and/or PI$^+$ cells) using flow cytometry. (D) Representative FACS plots from one healthy donor (HD) and one FMF patient are shown. Percentage are indicated for the two right gates. (E) Cell death modality was assessed by determining the percentage of treatment-induced Annexin-V$^+$/PI$^-$ cells and of PI$^+$ cells among dead cells, using flow cytometry.

Data information: (B, C, E) Kruskal–Wallis with Dunn's multiple comparison tests were performed to compare HD and FMF responses. Adjusted *P*-values are detailed below. (B) Each dot (HD)/symbol (FMF) represents the percentage of cells containing an ASC speck for one individual. Symbol to FMF patient #: square #2 (M694I/M694I), round #26 (V726A/V726A), triangle #13 (M694V/R761H), diamond #18 (M694I/M694I). UCN-01 **$P = 0.0095$; LPS + nigericin N.S. $P = 0.75$. (C) Each dot (HD)/symbol(FMF) represents the percentage of cells stained with FLICA-Casp1 for one individual, the bar represents the median ($\pm$interquartile range). Symbol to FMF patient #: round #26 (V726A/V726A), triangle #13 (M694V/R761H), triangle pointing down #23 (M694V/M694V), hexagon #24 (M694V/M694V), star #10 (M694V/M694V). Untreated: N.S. $P = 1$; UCN-01: **$P = 0.0069$; LPS + nigericin: N.S. $P = 1$. (E) Each dot (HD)/triangle (FMF) represents the value for one individual, and the bar represents the median ($\pm$ interquartile range). UCN-01: ***$P < 0.0001$; LPS + nigericin N.S. $P = 0.51$.
Source data are available online for this figure.

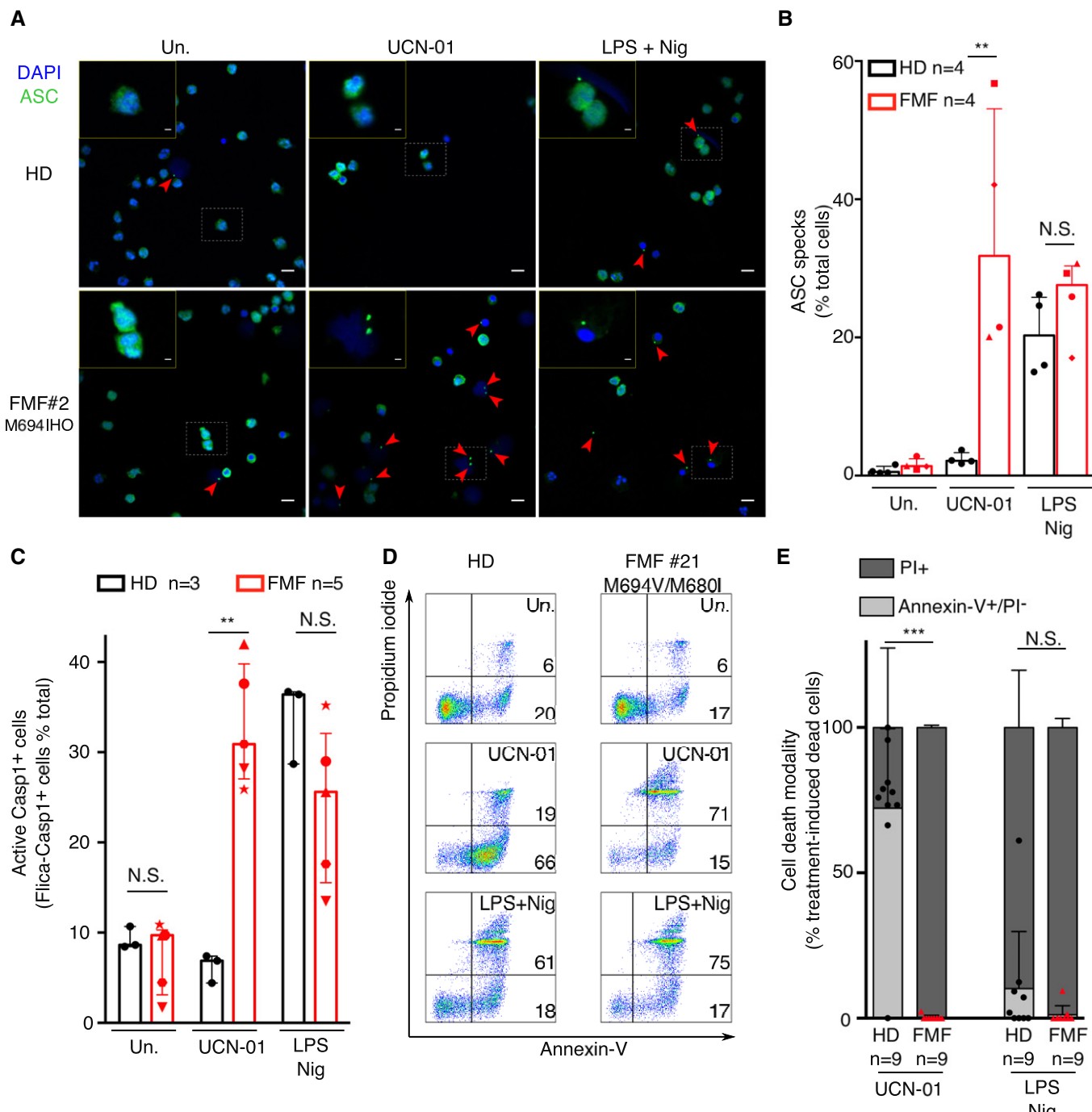

**Figure 2.**

## Inflammasome activation in FMF patients upon PKC inhibition is blocked by colchicine

The link between Pyrin and NLRP3 inflammasomes in FMF patients is still unclear (Chae et al, 2011; Omenetti et al, 2014). We thus tested whether NLRP3 could contribute to UCN-01-mediated inflammasome activation. As expected, MCC950-mediated inhibition of the NLRP3 inflammasome (Coll et al, 2015) abolished IL-1β release upon LPS + nigericin treatment (Fig 3A). MCC950 treatment

did not affect IL-1β release by FMF patients' monocytes exposed to TcdB in line with previous results obtained in PBMCs from HD exposed to TcdA (Van Gorp et al, 2016). In addition, MCC950 treatment did not affect IL-1β release upon UCN-01 treatment, indicating that this response is independent of NLRP3 (Fig 3A and Appendix Fig S5).

Colchicine specifically blocks the Pyrin inflammasome in murine macrophages and in PBMCs from healthy donors (Gao et al, 2016; Park et al, 2016; Van Gorp et al, 2016). Despite its long

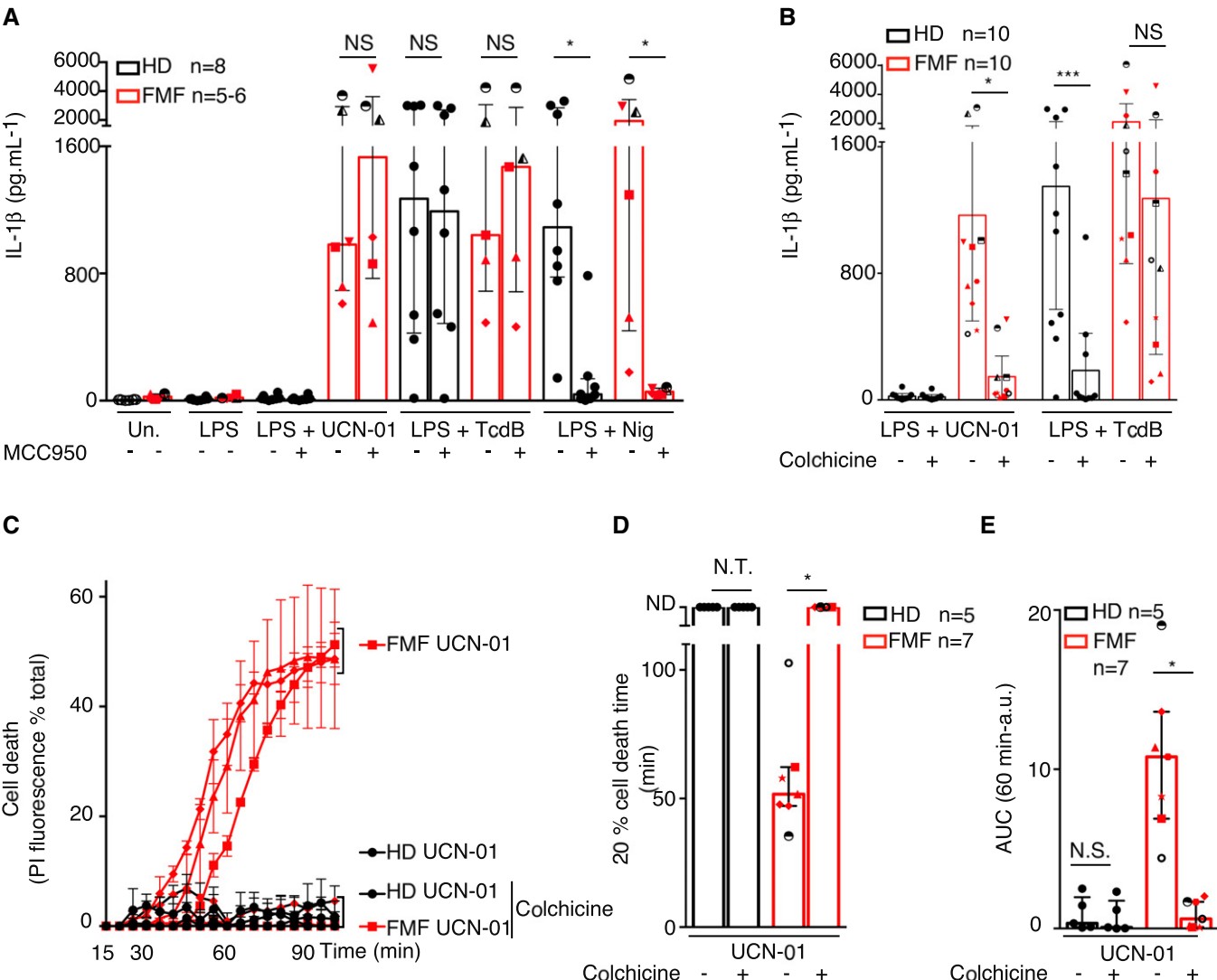

**Figure 3. Colchicine blocks inflammasome activation in monocytes from FMF patients following PKC inhibitor treatment.**

Monocytes from healthy donors (HD) or FMF patients were primed for 3 h as indicated with LPS (A, B) or not (C–E). When indicated, (A) the NLRP3 inhibitor MCC950 (10 μM) or (B–E) colchicine (1 μM) were added 30 min before addition of 12.5 μM UCN-01, 125 ng/ml TcdB, or 5 μM nigericin.

A, B  IL-1β level was quantified by ELISA at 90 min post-stimulation.

C–E  Cell death was monitored in real time by measuring propidium iodide influx/fluorescence every 5 min.

Data information: (A, B, D, E) Each symbol represents the mean value from three biological replicates for one HD or patient. The bar represents the median value (± interquartile range). (C) Each point of the curve corresponds to the average cell death values from three biological replicates of monocytes from HD or FMF patients (diamond #32 (p.M694V/p.M694V); triangle #33 (p.M694V/p.M694V); square #1 (p.M694I/p.V726A). The curves are displayed for each patient/HD. The time required to reach 20% cell death (D) and the area under the curve (AUC) (E) were computed for each HD or FMF patients. Each symbol represents the mean value from one HD or patient. The bar represents the median (± interquartile range). (D) ND indicates that the average cell death was below 20% at the end of the kinetics. (A) Wilcoxon matched-pairs signed rank test was used to compare untreated and MCC950-treated groups since all the groups did not contain an identical number of individuals. FMF, UCN-01 NS : P = 0.84, HD TcdB NS: P = 0.81, FMF TcdB NS: P = 0.84, HD Nig *P = 0.039, FMF Nig *P = 0.031. Individual genotypes are presented in Appendix Figure S5A. (B) Friedman paired test with Dunn's correction for multiple comparisons was applied to compare untreated and colchicine-treated groups. Adjusted P-values are as follow: FMF UCN-01 *P = 0.036; HD TcdB ***P = 0.0006; and FMF TcdB NS P = 0.28. (D, E) Wilcoxon matched-pairs signed rank test was used to compare untreated and colchicine-treated groups. Adjusted P-values are as follow: (D) N.T.: not tested. FMF *P = 0.016; (E) HD N.S. P = 0.62, FMF *P = 0.016. Source data are available online for this figure.

demonstrated clinical efficacy in FMF patients, colchicine was recently shown to be inefficient in blocking IL-1β release in PBMCs from FMF patients exposed to TcdA (Van Gorp *et al*, 2016). Although we did see a partial (40%) inhibition of TcdB-mediated IL-1β response in monocytes from FMF patients, with a large inter-patient variability, this inhibition was consistent and almost total (86%) in HD monocytes, thus confirming that toxin-mediated Pyrin inflammasome activation is less sensitive to colchicine inhibition in FMF patients than in HD monocytes (Fig 3B and Appendix Fig S5). In contrast to the lack of (Van Gorp *et al*, 2016)/

partial (our result) inhibition observed upon TcdA/B-mediated Pyrin inflammasome activation, colchicine strongly reduced (86%) IL-1β release and fully abolished pyroptosis upon PKC inhibition (Fig 3B–E). Similarly, and in line with previous work using TcdA to stimulate the Pyrin inflammasome (Van Gorp *et al*, 2016), nocodazole abolished UCN-01-mediated IL-1β release, and cell death (Fig EV1A and B). No substantial reduction was observed when using Taxol (Fig EV1A and B). Colchicine decreased UCN-01-mediated IL-1β release from FMF monocytes (Fig EV1C) and TcdB-mediated IL-1β release from HD monocytes (Fig EV1D) with a similar dose response. This result suggests that *C. difficile* toxins TcdA/B and PKC superfamily inhibitors differentially affect Pyrin inflammasome activation in FMF patients' monocytes. Based on the efficacy of colchicine in FMF patients, it is tempting to speculate that PKC inhibitors better mimic the endogenous stimuli triggering Pyrin inflammasome during inflammatory flares.

## Expression of p.M694V MEFV is necessary and sufficient to trigger caspase-1- and gasdermin D-dependent responses to PKC inhibitors

To demonstrate that the difference in PKC inhibitor responses in monocytes from FMF patients and HD was specifically due to *MEFV* mutation, we generated U937 cells expressing either WT *MEFV* or p.M694V *MEFV*. U937 were invalidated for the *MEFV* gene (Lagrange *et al*, 2018) to avoid any possible confounding factor and complemented with either 3xFLAG-WT (Lagrange *et al*, 2018) or 3xFLAG-p.M694V *MEFV* under the control of a doxycycline-inducible promoter (Fig EV2A). The Pyrin immunoblot pattern obtained upon doxycycline addition was similar to the pattern previously described in PBMCs (Chae *et al*, 2008) with a major cleavage band around 50 kDa (Fig EV2B). Importantly, doxycycline-mediated expression of p.M694V *MEFV* rendered U937 sensitive to UCN-01, as determined by their fast cell death, while the expression of WT *MEFV* did not (Fig 4A). As expected, in the absence of doxycycline, no cell death was observed during 3 h post-UCN-01 treatment. These results demonstrate that the expression of p.M694V Pyrin is necessary and sufficient for the fast cell death response to UCN-01. In contrast, the expression of WT or p.M694V Pyrin did not substantially affect the cell death response following NLRP3 stimulation (Fig 4B). Similar results were obtained with the two clearly pathogenic *MEFV* variants, p.M694I and p.M680I (Fig EV3A–H). In contrast, the expression of the variant of unknown significance, p.P369S, did not trigger such a response suggesting that it is either a non-pathogenic variant (in line with its higher frequency in the human population Fig EV3I), that its pathogenicity is undetectable in our experimental system, or that its pathogenicity is associated with another molecular mechanism.

Importantly, the UCN-01-mediated cell death was strongly delayed when p.M694V Pyrin was expressed in *CASP1* or *GSDMD* knock-out cells, indicating that UCN-01 triggers pyroptosis in p.M694V Pyrin-expressing cells (Fig 4C). Invalidation of *GSDMD* did not impact cell death kinetics as much as *CASP1* invalidation. Such differences in cell death kinetics have been reported in several studies comparing cell death kinetics upon invalidation of individual inflammasome components and likely reflect the existence of alternative/secondary cell death pathways (Pierini *et al*, 2012; Sagulenko *et al*, 2013; Schneider *et al*, 2017).

Doxycycline-mediated expression of p.M694V Pyrin was necessary and sufficient to trigger a robust production of IL-1β (Fig 4D) and IL-18 (Fig EV2C) in response to PKC inhibitors. Once again, the differences between WT and p.M694V Pyrin-expressing cell lines in terms of IL-1β secretion were largely specific for PKC inhibitors, although low levels of IL-1β and IL-18 secretion were observed in p.M694V Pyrin-expressing cells in the absence of PKC inhibitors, possibly due to the use of PMA (Figs 4D and EV2C). UCN-01/p.M694V Pyrin-mediated IL-1β was dependent on both *CASP1* and *GSDMD* (Fig 4E), indicating that this response is a bona-fide inflammasome response.

UCN-01 is a PKC superfamily inhibitor and thereby not a specific inhibitor of PKN1/2. We were unable to invalidate *PKN2* by CRISPR/Cas9 despite multiple assays and more than 50 other human genes successfully invalidated in the meantime in the same cellular system (Benaoudia *et al*, 2019). *PKN2*-edited clones were recovered but contained small deletions/insertions not affecting the open-reading frames (see one example in Appendix Fig S7), suggesting that PKN2 is necessary for cell growth/survival of U937 cells. This observation is in line with the embryonic lethality of *Pkn2* knock-out in mice (Danno *et al*, 2017). We thus generated *PKN1*[KO] clones expressing either WT or p.M694V Pyrin (Fig EV4A) and performed siRNA-mediated knockdown (KD) of *PKN2*. We were unable to detect PKN2 protein by Western blot analysis likely due to its low expression in U937 cells (Fig EV4B). Yet, two (siRNA #12 and 13) out of three siRNA gave us a strong reduction in *PKN2* transcript levels (Fig EV4C). Doxycycline-mediated induction of p.M694V Pyrin in *PKN1*[KO] *PKN2*[KD] cells increased cell death levels (Fig EV4D–E) compared to (i) cells not expressing Pyrin, (ii) cells expressing WT Pyrin, and (iii) p.M694V-expressing cells treated with a non-targeting (NT) siRNA or an inefficient *PKN2*-targeting siRNA (siRNA #14, Fig EV4F). Although the cell death levels were low, these results suggest that *PKN1/2* genetic invalidation mimics UCN-01 treatment and, in the presence of p.M694V Pyrin, is sufficient to trigger cell death.

## PKC superfamily inhibitors regulate serine 242 phosphorylation to trigger cell death in p.M694V-expressing cells

To validate that the absence of inflammasome activation in WT Pyrin-expressing U937 cells was not due to a lack of UCN-01 efficacy and a lack of Pyrin dephosphorylation, we assessed the (de)phosphorylation of Pyrin Ser242 by Western blot analysis. The specificity of the antibody (Gao *et al*, 2016) was validated in our experimental system using a cell line expressing p.S242R Pyrin (Appendix Fig S8). Importantly, we observed dephosphorylation of Pyrin Ser242 residue upon UCN-01 treatment in WT Pyrin-expressing cells (Fig 4F) that was similar to the dephosphorylation pattern observed in p.M694V Pyrin-expressing cells.

Since the dephosphorylation of Pyrin only leads to pyroptosis in the presence of the p.M964V mutation, we reasoned that the p.S242R mutation, which blocks phosphorylation of Pyrin on this key residue, should be cytotoxic in the presence of the p.M694V mutation. Indeed, induction of the expression of a Pyrin protein containing both the p.S242R and the p.M694V mutations in U937 cells was cytotoxic per se in the absence of PKC superfamily inhibitors (Fig 5A and C). Similarly, induction of the expression of a Pyrin protein expressing the p.S208C mutation and the p.M694V mutations was also cytotoxic (Fig 5A and C). Cell death upon induction

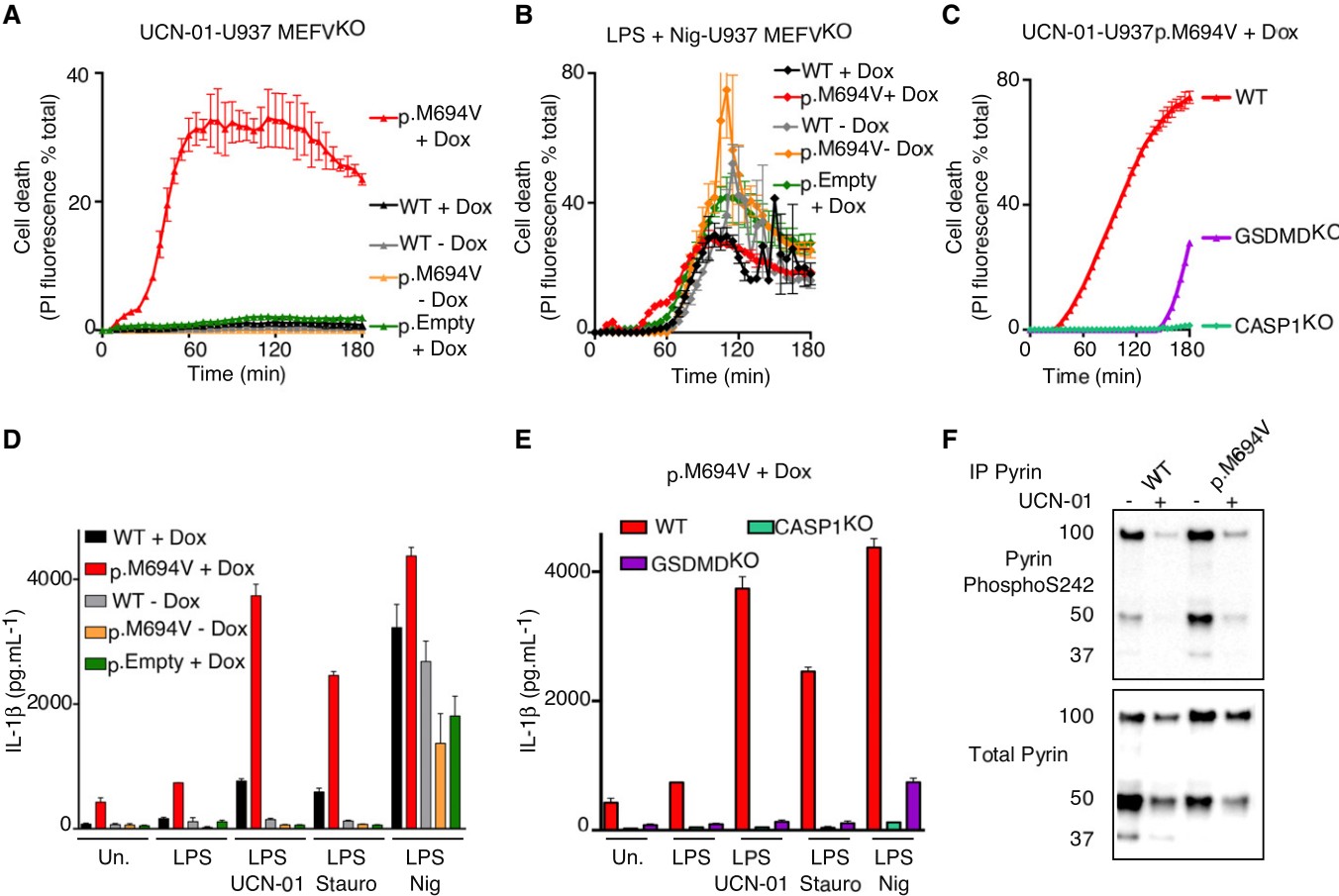

**Figure 4. Expression of p.M694V Pyrin is necessary and sufficient to confer to PKC inhibitors the ability to activate the inflammasome.**

A–C U937 cells of the indicated genotype and expressing the indicated plasmids were treated with (A, C) UCN-01 or LPS + nigericin (Nig). Propidium iodide (PI) influx/ fluorescence was monitored every 5 min for 3 h.

D, E PMA-differentiated U937 cells of the indicated genotype and expressing the indicated plasmids were primed with LPS for 3 h and treated with UCN-01, staurosporine (Stauro) or nigericin as indicated. IL-1β level in the supernatant was quantified by ELISA at 3 h post-treatment.

F Pyrin S242 phosphorylation was assessed by Western blot in the indicated cell lines with and without UCN-01 treatment for 15 min.

Data information: (A–C) One experiment representative of three independent experiments is shown. Mean and standard deviations from two biological replicates are shown. (D, E) Mean and standard deviations from three independent experiments are shown.

Source data are available online for this figure.

of these double mutants was reproducibly observed although the maximal cell death levels were variable between experiments (mean = 36.7% ± 20.1(SE) [min: 11.2, max 67.1; $n = 7$] for p.S242R/M694V and mean = 24.4% ± 6.7 (SE) [min 15.8, max 27.9; $n = 4$] for p.S208C/M694V). Interestingly, the level of cell death observed in cells expressing a Pyrin protein containing the two phospho-null mutations and the p.M694V mutations (triple mutant p.S208C/S242R/M694V) was consistently higher than the one of the cells expressing a single phospho-null mutation in combination with the p.M694V mutation (Fig 5B and C). The cell death kinetics correlated with the kinetics of protein expression following doxycycline-mediated induction (Fig EV2A and B, and Appendix Fig S9). In contrast, neither expression of the single or double phospho-null mutant (p.S208C; pS242R; p.S208C/S242R) nor of the p.M694V single mutant proteins triggered substantial cell death. Addition of UCN-01 at 20 h post-doxycycline addition had a minor additional cytotoxic effect on the p.S208C/M694V and the

p.S242R/M694V-expressing cell lines (Fig 5A and D). No additional cytotoxic effect could be detected on the p.S208C/S242R/ M694V-expressing cell line since all the cells were dead at 20 h post-doxycycline addition. Interestingly, colchicine addition could block the cell death induced by the doxycycline-mediated expression of pS208C/M694V, pS242R/M694V, and pS208C/S242R/ M694V Pyrin variants (Fig EV5), suggesting that colchicine acts independently (and downstream; Van Gorp et al, 2016; Gao et al, 2016) of Pyrin dephosphorylation.

Similarly, doxycycline-mediated induction of p.S208C/M694V or p.S242R/M694V triggered IL-18 release in the absence of PKC inhibitors (Fig 5E). When added onto the p.M694V background, the two phospho-null mutations had an additive effect in regard to IL-18 concentration in the supernatant (Fig 5E). As described above for the cell death, IL-18 was not substantially released upon doxycycline induction of p.M694V, p.S208C, p.S242R, or p.S208C/S242R Pyrin proteins (Fig 5E). As expected, a further addition of UCN-01

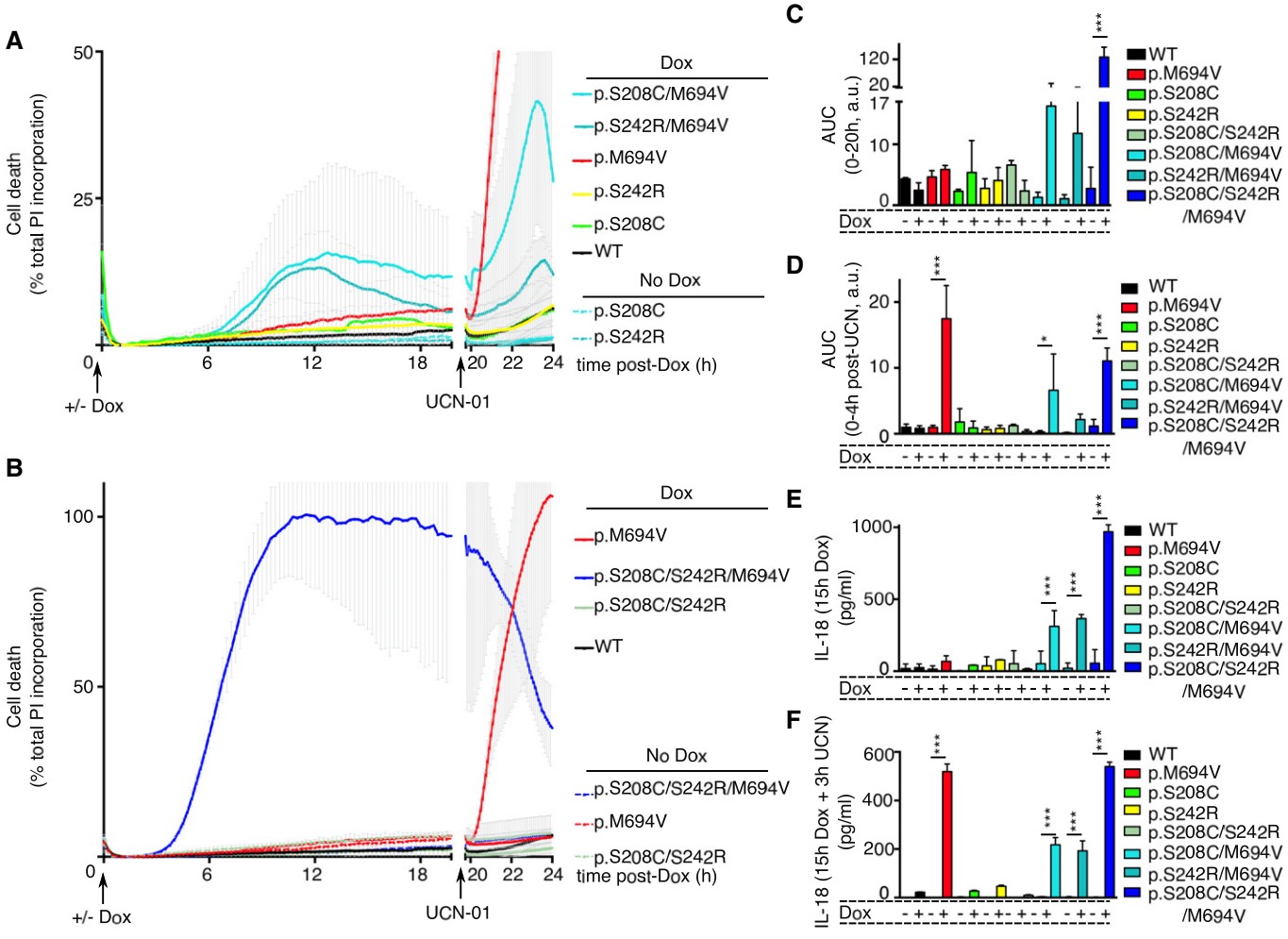

**Figure 5. Expression of p.M694V Pyrin is necessary and sufficient to confer to the phospho-null p.S242R/p.S208C variants the ability to activate the inflammasome.**

U937 cells bearing the indicated plasmids were treated as indicated at time 0 with doxycycline (Dox) and at 20 h post-Dox addition with UCN-01

A, B Propidium iodide (PI) influx/fluorescence was monitored every 15 min for 20 h and every 5 min for an additional 4 h.

C, D Area under the curve (AUC) corresponding to Fig 5A and B was computed from $t = 0$ to $t = 20$ h (C) and from 20 to 24 h (D).

E, F IL-18 was quantified in the supernatant of the indicated U937 cells at 15 h post-doxycycline/PBS addition (E) or in the supernatant of U937 cells exposed to 15 h of doxycycline/PBS and an additional 3 h of incubation with UCN-01 (F).

Data information: (A–F) One experiment representative of three independent experiments is shown. (A, B) Each dot represents the mean ± SD of a biological triplicate. Values right after UCN-01 addition were corrected to match values right before UCN-01 addition to correct for a reading artifact, and the correction factor was applied until the end of the experiment. Data presented in (A and B) are from the same experiment and have been split for clarity. The WT and p.M694V controls are duplicated in (A and B). (C–F) The bar represents the mean ± SD of a biological triplicate. One-way ANOVA with Holm–Sidak's multiple comparisons test was performed to compare untreated and doxycycline-treated samples: (C) p.S208C/S242R/M694V ***$P < 0.0001$ (D) p.M694V ***$P < 0.0001$; p.208C/M694V *$P = 0.016$; p.S208C/S242R/M694V ***$P = 0.0004$. (E) p.208C/M694V ***$P = 0.0001$; p.S242R/M694V ***$P < 0.0001$; p.S208C/S242R/M694V ***$P < 0.0001$. (F) p.M694V ***$P < 0.0001$; p.208C/M694V ***$P < 0.0001$; p.S242R/M694V ***$P < 0.0001$; p.S208C/S242R/M694V ***$P < 0.0001$.

Source data are available online for this figure.

triggered the fast release of IL-18 in p.M694V-expressing cells while no IL-18 was observed in cells expressing only the phospho-null Pyrin mutants (Fig 5F).

These data provide genetic evidence that dephosphorylation of Pyrin (or lack of phosphorylation) in the context of the p.M694V mutation is necessary and sufficient to promote full Pyrin inflammasome activation. Our data suggest that dephosphorylation of the S208 and of the S242 residues have an additive (IL-18) or possibly synergistic (cell death) effect to trigger Pyrin inflammasome

activation in the context of p.M694V mutation. Further, these results strongly suggest that UCN-01 acts on both S242 and S208 to trigger the fast cell death observed in primary monocytes from FMF patients and in U937 cells expressing the p.M694V Pyrin variant. Dephosphorylation of p.M694V Pyrin is thus sufficient to trigger inflammasome activation while another safeguard mechanism exists to control activation of dephosphorylated WT Pyrin. Finally, these results provide the proof of concept that U937 cells and PKC inhibitors could be used to functionally characterize *MEFV* variants for their pathogenicity.

**PKC inhibitor responses functionally discriminate FMF patients from patients with unrelated inflammatory conditions**

To exclude that the response of monocytes from FMF patients might be due to inflammation, we then studied the responses of monocytes from patients presenting various inflammatory syndromes and infection-associated inflammation including Behcet's disease, inflammatory bowel disease, and sepsis (Appendix Table S1). While each disease was represented by a low number of patients, none of the non-FMF patients (termed "Disease control-DC" in Fig 6) responded to the two PKC inhibitors by producing large amount of IL-1β (Fig 6A and B) or by triggering a fast monocyte death (Fig 6C–E, Appendix Fig S10A–C). Even though larger cohorts of patients with "control diseases" are required to validate statistically this difference for each inflammatory syndrome, these data strongly suggest that inflammasome activation in response to PKC inhibitors is not due to underlying inflammation but is specific to FMF patients.

Due to this discrimination between FMF patients and other patients suffering from various conditions with an inflammatory component, we assessed whether the functional response to PKC inhibitors has the potential to be exploited for FMF diagnosis. We thus generated receiver operating characteristic (ROC) curves to determine the sensitivity and specificity of a functional test based on IL-1β release following staurosporine (Fig 6F) or UCN-01 treatment (Fig 6G) or based on cell death kinetics parameters (time to reach 20% cell death; Fig 6H or AUC of the cell death kinetics; Fig 6I). The predictive values (Appendix Table S2) and the areas under the ROC curves, which are very close to 1 (1 corresponding to 100% specificity and 100% sensitivity), suggest that these functional assays accurately discriminate FMF patients from other patients presenting inflammatory conditions and from HD (Appendix Fig S10D–G). As an example, and keeping in mind that such a diagnostic use remains to be thoroughly tested in larger and multicentric cohorts, an IL-1β concentration threshold at 224 pg/ml in response to UCN-01 discriminates FMF patients with a sensitivity of 89% and a specificity of 96%. Interestingly, the tests based on the cell death kinetics have slightly better positive and negative predictive values (Appendix Table S2) than the tests based on IL-1β, suggesting that pyroptosis monitoring better discriminates FMF patients from HD (Appendix Fig S10F–G) and from patients with other inflammatory conditions (Fig 6H and I) than quantifying IL-1β release does.

In conclusion, our study demonstrates that PKC inhibitor treatment is sufficient to trigger inflammasome activation in monocytes from FMF patients, which (i) expands our knowledge on the molecular basis sustaining Pyrin inflammasome deregulation in FMF, (ii) offers a potential functional assay to characterize *MEFV* variants, and (iii) provides a fast biological test that may discriminate FMF patients from patients suffering from other inflammatory conditions.

# Discussion

The links between *MEFV* mutations, the dysfunctional Pyrin inflammasome, and FMF remain poorly understood. In contrast to the typical gain-of-function mutations of *NLRP3* or *NLRC4* observed in cryopyrin-associated periodic syndromes and NLRC4-associated syndromes, *MEFV* mutations do not lead to a spontaneous activation of the Pyrin inflammasome in the absence of a specific stimulus

(Van Gorp *et al*, 2016; Jamilloux *et al*, 2018). We have previously shown that *MEFV* mutations lower the activation threshold of the Pyrin inflammasome in response to a Pyrin inflammasome stimulus (Jamilloux *et al*, 2018). While our previous work demonstrated a shift in a dose–response experiment between monocytes of FMF patients and of healthy donors, here we identified an "all or nothing" response that discriminates FMF patients' monocytes from healthy donors' monocytes across a large concentration of PKC superfamily inhibitors.

*MEFV* mutations were described to lift the obligatory requirement for microtubule in Pyrin inflammasome activation (Van Gorp *et al*, 2016). In the work by van Gorp and colleagues, colchicine was inefficient at inhibiting TcdA-mediated Pyrin inflammasome in FMF patients PBMCs. Here, we took a reverse approach not based on inhibition of the Pyrin inflammasome (by colchicine) but on activation of the Pyrin inflammasome, which we demonstrated to be specific for FMF patients' monocytes following PKC superfamily inhibitor addition. We believe our results are important since they clearly demonstrate that dephosphorylation of Pyrin is sufficient to promote full inflammasome activation in FMF patients while it is not sufficient in healthy donors.

Our results partially concur with the conclusions from van Gorp and colleagues that FMF mutations affect the ability of colchicine to inhibit Pyrin inflammasome activation. Yet, our results demonstrate that the phenotype of FMF monocytes (and likely FMF patients) cannot be explained by a full loss of microtubule-dependent activation. The interaction of *MEFV* mutations with this microtubule-dependent step is likely much more complex. While colchicine fully inhibited TcdB-mediated IL-1β in healthy donor monocytes, colchicine inhibited only partially IL-1β release in FMF patients' monocytes. Furthermore, colchicine fully inhibited pyroptosis and IL-1β release in response to PKC inhibition in FMF patients' monocytes. Similarly, colchicine inhibited cell death induced by the expression of a Pyrin protein containing both the p.S242R/S208C and the p.M694V mutations (Fig EV5). This *in vitro* efficacy of colchicine is in line with the efficacy of colchicine observed in FMF patients and with the previous *in vitro* assays performed on FMF monocytes following LPS stimulation (Park *et al*, 2016). One important message of our work is thus that colchicine is efficient at inhibiting the Pyrin inflammasome (at least in response to some stimuli) in FMF patients.

Our results demonstrate that, while the PKC superfamily inhibitors are efficient to dephosphorylate both WT Pyrin and p.M694V Pyrin (Park *et al*, 2016), ASC oligomerization, pyroptosis, and robust IL-1β release are only triggered when these inhibitors are added on cells expressing p.M694V *MEFV* gene. This study thus reinforces the data indicating that p.M694V mutation does not impact Pyrin phosphorylation status in a major way (Gao *et al*, 2016; Masters *et al*, 2016; Van Gorp *et al*, 2016), but affects another control mechanism, likely related to a complex process implicating microtubule dynamics (Gao *et al*, 2016; Van Gorp *et al*, 2016).

PAAND patients presenting the p.S242R mutation, which abolishes one Pyrin phosphorylation site (Masters *et al*, 2016) or patients presenting the p.E244K mutation, which abolishes 14-3-3 binding site (Moghaddas *et al*, 2017), have a constitutive activation of the Pyrin inflammasome. Yet, the level of IL-1β released by PAAND patients' monocytes upon LPS stimulation is moderate (Masters *et al*, 2016) and at least 10 times lower than the level of IL-1β released upon stimulation with LPS + TcdB. This observation suggests that in PAAND

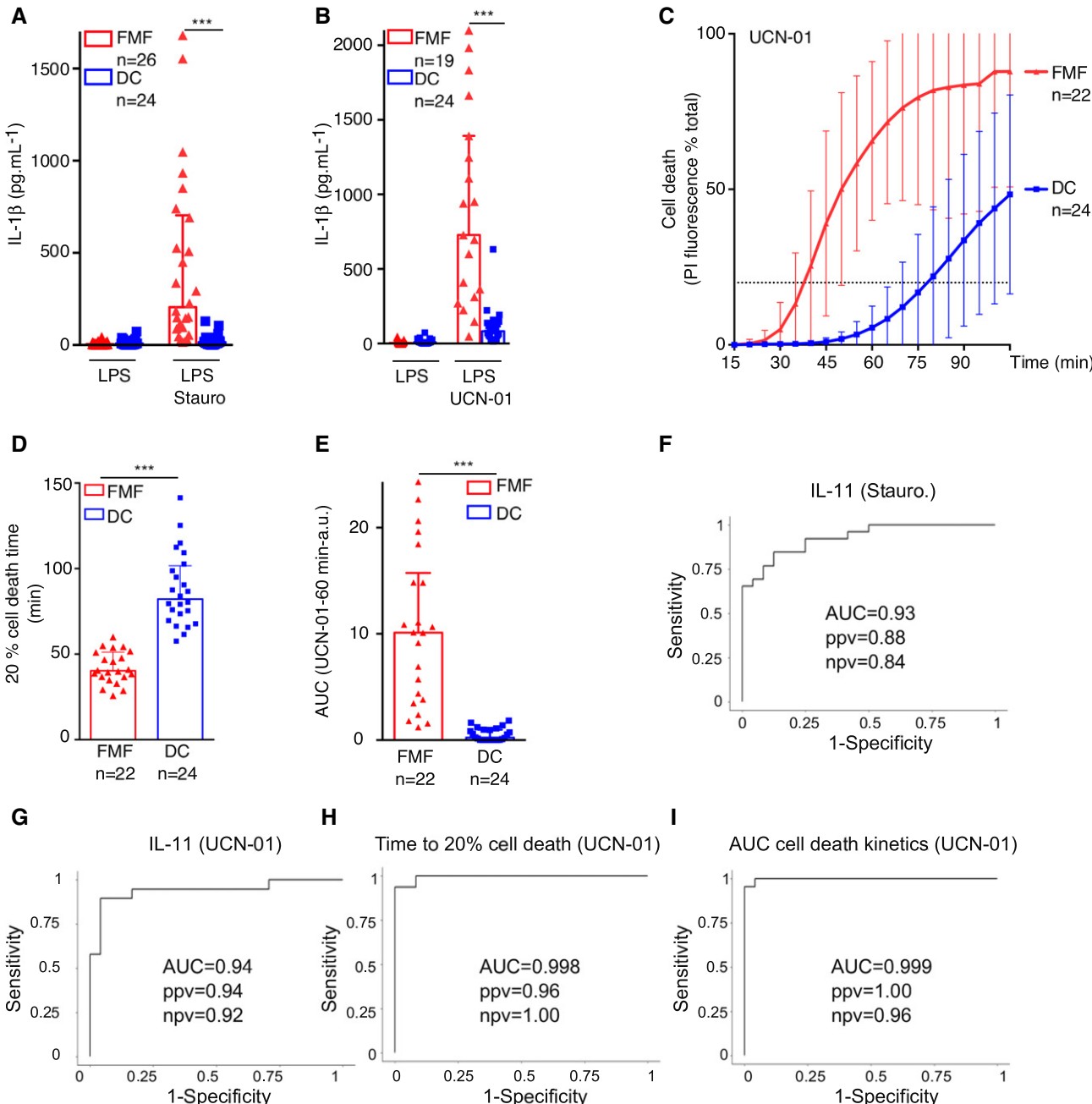

**Figure 6. PKC inhibitor-mediated inflammasome activation discriminates FMF patients from patients suffering from unrelated inflammatory conditions.**

Monocytes from FMF patients or from control diseases patients (DC) were either primed with LPS (A, B) or not (C–E) and treated with (A) 1.25 µM staurosporine (Stauro), (B, C–E) 12.5 µM UCN-01.

A, B   IL-1β level was quantified by ELISA at 90 min post-stimulation.

C–E   Cell death was monitored in real time by measuring propidium iodide (PI) influx/fluorescence every 5 min. (D) The time required to obtain 20% of cell death and (E) the area under the kinetics curve (AUC) were computed for each patient.

F–I   Receiver operating characteristic (ROC) curves were computed for IL-1β concentrations following (F) staurosporine or (G) UCN-01 treatment, (H) the time to obtain 20% cell death, and (I) the area under the cell death kinetics curve. For each ROC curve, the AUC, the positive (ppv), and negative (npv) predictive values are indicated.

Data information: (A, B) Each symbol represents the mean value from three biological replicates for one patient. The bar represents the median ± interquartile range. (C) Each point of the curve corresponds to the average of the mean cell death values from three biological replicates of monocytes from the indicated number of patients. The dotted line indicates the 20% cell death value. (D–E) Each dot represents the value from one patient. The bar represents the median ± interquartile range. (A, B): ***P < 0.0001 by Wilcoxon rank-sum test. (D, E) ***P < 0.0001 by unpaired t-test. (A–E) Values from FMF patients are identical as the ones presented in Fig 1. The figures were not merged for clarity issues.

Source data are available online for this figure.

patients' monocytes, an additional control mechanism independent of phosphorylation/14-3-3-binding still partially limits Pyrin inflammasome activation. In agreement with this hypothesis, ectopic expression of a double mutant p.S242R/p.M694V Pyrin protein in HEK293T cells was found to promote more GFP-ASC speck formation than a single p.S242R Pyrin protein (Masters *et al*, 2016). Similarly, controlled expression of phosphorylation-mutant Pyrin in the mouse dendritic cell line DC2.4 was reported to trigger ≈ 20% cell death (Gao *et al*, 2016). We did not observe substantial cell death in U937 cells expressing p.S242R (or p.S208C/S242R) Pyrin in contrast to the cytotoxicity associated with the expression of p.S242R/M694V double mutant (or p.S208C/S242R/M694V triple mutant). All together, our results combined with previous data from the literature emphasize that the WT Pyrin inflammasome is controlled by two largely independent mechanisms. In contrast, in FMF patients, the Pyrin inflammasome activation is solely under the control of PKC superfamily kinases and lacks a second negative control mechanism likely explaining its hyper-reactivity and the associated recurrent inflammation observed in FMF patients. Similarly, in PAAND patients where the phosphorylation regulation is affected, we believe the Pyrin inflammasome is not fully activated due to the control by the second colchicine-targetable mechanism. Yet, clinical evidence demonstrates that missing one of these two control mechanisms is associated with disease highlighting the key requirement for a tight inflammasome control. A key challenge in the field is now to identify the factors that could mimic PKC superfamily inhibitors and be responsible for inflammatory flares in patients.

In the present study, most FMF patients were under colchicine treatment (Appendix Table S1). Colchicine treatment is unlikely to explain the hyper-reactivity of FMF patients to PKC superfamily inhibitors since (i) we could recapitulate the findings in U937 cells; (ii) *in vitro*, colchicine reduces inflammasome activation (Gao *et al*, 2016; Park *et al*, 2016; Van Gorp *et al*, 2016); and (iii) three patients with Behcet's disease under colchicine were included in the disease control group and their monocytes did not display an FMF-like phenotype. Although our data suggest that a single clearly pathogenic *MEFV* variant is sufficient to confer to PKC inhibitors the ability to trigger inflammasome activation, our cohort of FMF patients is currently too small to draw robust conclusions regarding the number (biallelic mutations vs. mono-allelic) of clearly pathogenic variants. Furthermore, due to the large number of *MEFV* variants present in our cohort, conclusions on specific *MEFV* variants will require the recruitment of a large number of FMF patients coupled to the functional characterization of *MEFV* variants in genetically engineered cell lines.

Here, we show that the response of monocytes from FMF patients to PKC inhibitors could discriminate FMF patients from other patients presenting inflammatory conditions. Our results thus pave the way toward the development of a functional diagnostic test for FMF. Of note, several limitations of the current study remain to be overcome to fully evaluate the diagnostic potential of this assay. First, the sensitivity of the test has to be defined with respect to the different *MEFV* genotypes. Second, the assay would have to be validated in whole blood to be compatible with clinical laboratory. Third, the test remains to be validated in a large independent, multicentric cohort of patients in a prospective manner. Such a test could be used upstream of genetic tests to narrow-down the diagnosis or downstream of the genetic test in the case of non-conclusive results, which represent about 25% of the current genetic diagnosis

(Grateau *et al*, 2000; Lachmann *et al*, 2006; Booty *et al*, 2009; Giancane *et al*, 2015). Furthermore, a functional assay might help physicians to take decisions regarding the weaning of the lifelong colchicine treatment in FMF patients possessing a single *MEFV* pathogenic variant (Sonmez *et al*, 2017). The lack of colchicine inhibition of the Pyrin inflammasome in response to TcdA can also be used to discriminate FMF patients from HD (Van Gorp *et al*, 2016). Interestingly, our functional test does not require the manipulation of the highly potent *C. difficile* toxins and might be thus easier to implement in the hospital/diagnostic laboratory environment. The *in vitro* efficacy of colchicine could thus be combined to the responsiveness to PKC inhibitors to further increase the potential of functional assays to discriminate HD and FMF patients.

## Materials and Methods

### Subjects

Thirty-nine patients with FMF (Appendix Table S1) were included along with patients suffering from other inflammatory conditions (mevalonate kinase deficiency ($n = 3$), Behcet's disease ($n = 4$), lupus ($n = 2$), Still's disease ($n = 5$), non-systemic juvenile idiopathic arthritis ($n = 1$), sepsis ($n = 3$), inflammatory bowel disease ($n = 4$), H syndrome ($n = 1$), A20 deficiency ($n = 1$)), and 45 HD. All FMF patients fulfilled the Tel HaShomer criteria for FMF and had at least one mutation in the *MEFV* gene. The potential carriage of *MEFV* mutations in HD was not assessed. Blood samples from HD were drawn on the same day as patients. Allele frequency was extracted from https://gnomad.broadinstitute.org/ (Lek *et al*, 2016).

### Monocyte isolation

Blood was drawn in heparin-coated tubes and kept at room temperature overnight. The next day, peripheral blood mononuclear cells (PBMCs) were isolated by density-gradient centrifugation using lymphocyte separation medium (Eurobio; Noble & Cutts, 1968). Monocytes were isolated from PBMCs by magnetic selection using CD14 MicroBeads (Miltenyi Biotec) (Lyons *et al*, 2007) and the AutoMACS Pro Separator (Miltenyi Biotec) following manufacturer's instructions. Monocytes were enumerated in the presence of a viability marker (propidium iodide, 10 μg/ml) by flow cytometry (BD Accuri C6 Flow Cytometer®) (Schuerwegh *et al*, 2001).

### Inflammasome activation

Primary monocytes were seeded into 96-well plates at $5 \times 10^3$ cells/well, in RPMI 1640, GlutaMAX medium (Thermo Fisher Scientific) supplemented with 10% fetal calf serum (Lonza). When indicated, primary monocytes were incubated for 3 h in the presence of LPS (10 ng/ml, InvivoGen). Unless otherwise indicated, primary monocytes were then treated for 90 min with nigericin (5 μM, InvivoGen), ATP (2.5 mM, Sigma) (Mariathasan *et al*, 2006), staurosporine (1.25 μM, Tocris), UCN-01 (12.5 μM, Sigma), or Ro31-8820 (100 μM, Tocris). When indicated, monocytes were treated with colchicine (1 μM or at the indicated concentration, Sigma), MCC950 (10 μM, Adipogen AG-CR1-3615), paclitaxel (Taxol, 5 μM, Sigma), nocodazole (5 μM, Sigma), z-YVAD-FMK, z-IETD-FMK, z-DEVD-FMK (at the

indicated concentrations, Bachem, #4027532, #4034771, #4027402, respectively), VX-765 (InvivoGen), 30 min before addition of UCN-01, TcdB (Abcam, #ab124001, 125 ng/ml), and TcdA (1 μg/ml) or niger-icin. TcdA was purified from *Clostridioides difficile* VPI10463 strain, as previously described (von Eichel-Streiber *et al*, 1987; Popoff, 1987). Following the incubation, cells were centrifuged and supernatants were collected.

**Immunofluorescence**

Monocytes ($1.2 \times 10^5$ per sample) stimulated as indicated above were fixed with paraformaldehyde 2% for 20 min at 37°C, washed three times in PBS, and spread onto poly-lysine adhesion slides (Thermo Fisher Scientific™) using the Cytospin 3 (Shandon) 5 min at 450 rpm. Following permeabilization with Triton X-100 (0.1% in PBS) and overnight incubation in blocking buffer (PBS, 3% BSA), cells were stained using anti-ASC (Santa Cruz, sc22514R, 4 μg/ml, 1 h of room temperature), Alexa 488-goat anti-rabbit antibodies (Invitrogen, A-110088, 10 μg/ml, 20 min of room temperature), and DAPI (100 ng/ml) all diluted in a 50 μl drop of blocking buffer. Coverslips were mounted onto the slides using mowiol. ASC specks and DAPI stained nuclei were visualized on the Zeiss LSM800 confocal microscope. Quantification was performed on 10 fields per sample.

**FLICA-Casp1 and Annexin-V staining**

Monocytes were stimulated as described above and stained with the FAM FLICA-Caspase-1 Kit (Bio-Rad ICT098), at the indicated time points, following manufacturer's instructions. Cells were extempora-neously analyzed on a Canto II cytometer. Annexin-V staining was performed using the Annexin-V Apoptosis Detection Kit (Invitrogen, 88-8005-74). Cells were then stained with propidium iodide (5 μg/ml) before acquisition on a CantoII cytometer. To obtain the response induced by the treatment, values from untreated samples were subtracted from values obtained following treatment. The sum of Annexin-V$^+$ and PI$^+$ cells defined the total dead cells. The ratio was calculated as the % of Annexin-V$^+$/PI$^-$ over total dead cells.

**Cell lines and genetic manipulation**

The human myeloid cell line U937 (CelluloNet, Lyon, France) was grown in RPMI 1640 medium with glutaMAX-I supplemented with 10% (vol/vol) FCS, 2 mM L-glutamine, 100 IU/ml penicillin, and 100 μg/ml streptomycin (Thermo Fisher Scientific). $MEFV^{KO}$, $Casp1^{KO}$, and $GSDMD^{KO}$ cell lines generated by CRISPR/Cas9-mediated gene invalidations have been previously described (Lagrange *et al*, 2018). The sgRNA targeting *PKN1* and *PKN2* (Appendix Table S3) were selected from the Brunello library (Addgene) and cloned into the PGKpuro2ABFP vector (from Kosuke Yusa; Addgene plasmid # 50946). sgRNA plasmids were transduced in a previously described Cas9-expressing U937 clone (Lagrange *et al*, 2018) by spinoculation. The resulting cell lines were selected with 2 μg/ml Puromycin (Sigma-Aldrich) at 72 h post-transduction for 2 weeks. Except for *PKN1* and *PKN2* invalidation (due to low to zero efficiency of the KO process), the knock-out cells were kept polyclonal and were screened by Western blotting analysis or sequencing of a PCR fragment corresponding to the genomic region flanking the targeted sequence. The obtained sequence files were analyzed using

the sequence trace decomposition software TIDE (Brinkman *et al*, 2014; Appendix Fig S7).

WT, p.M694V, p.S242R, and p.S242R/M694V *MEFV* were cloned into the GFP-expressing plasmid pINDUCER21 (Meerbrey *et al*, 2011) under the control of a doxycycline-inducible promoter through the pENTR1A (Invitrogen) vector using a synthetic DNA fragment (GeneArt) encoding the p.M694V Pyrin protein. The point mutants were generated using the QuikChange II Site-Directed Mutagenesis Kit (Agilent) and primers indicated in Appendix Table S3. Lentiviral parti-cles were produced in 293T cells using pMD2.G and psPAX2 (from Didier Trono, Addgene plasmids #12259 and #12260), and pINDUCER-21 plasmids expressing WT, p.M694V, p.S242R, p.S242R/M694V Pyrin, or no insert (pEmpty). The various U937 cell lines were trans-duced by spinoculation and sorted at day 7 post-transduction based on GFP expression on an Aria cell sorter. Pyrin expression was induced by treatment with doxycycline (1 μg/ml) for 16 h before stimulation. For real-time cell death assay, U937 cells were seeded at $4 \times 10^4$ per wells before stimulation with relevant inflammasome stimuli. For IL-18 ELISA, $2 \times 10^5$ U937 cells per wells were seeded. To assess IL-1β release, $8 \times 10^4$ U937 cells per wells were exposed to 100 ng/ml of phorbol 12-myristate 13-acetate (PMA; InvivoGen) for 48 h and primed with LPS at 1 μg/ml for 3 h. When applicable, nigericin was used at 50 μg/ml. Supernatant was collected at 3 h post-treatment. 293T and U937 parental cell lines were tested negative for mycoplasma contamination (Cellulonet, Lyon, France) in January 2018.

SiRNA targeting *PKN2* (ON TARGET plus, Dharmacon, # J0004612-12-0002; J0004612-13-0002; J0004612-14-0002) is indi-cated in Appendix Table S3. SiRNA was electroporated into U937 cell lines using the Neon electroporator (Thermo Fisher Scientific) using the following parameters: 1,300 V, 1 pulse of 30 ms. $10^6$ cells per condition were washed once at room temperature with PBS and suspended in 110 μl of buffer R containing 10 pmoles of siRNA. Following electroporation, cells were transferred into a well of a pre-warmed p12 plate containing complete medium without antibi-otics. When applicable, doxycycline was added at 24 h post-siRNA electroporation.

**Immunoprecipitation and immunoblot analysis**

Cells were lysed in 25 mM Tris–HCl, 150 mM NaCl, 1 mM EDTA, and 0.1% NP-40 buffer containing Mini Protease Inhibitor Mixture (Roche) and sodium fluoride (Sigma) by a quick freezing and thawing step. Flag-Pyrin was immuno-precipitated using anti-Flag M2 affinity gel (Sigma). Proteins were separated by SDS–PAGE on precast 4–15% acrylamide gels (Bio-Rad) and transferred to TransBlot® Turbo™ Midi-size PVDF membranes (Bio-Rad). Antibodies used were mouse monoclonal anti-FLAG® (Sigma-Aldrich, clone M2; 1:1,000 dilution), anti-Pyrin (Adipogen, AL196, 1: 1,000 dilution), anti-phospho S242 Pyrin (Abcam, ab200420; 1:1,000 dilution; Gao *et al*, 2016), and anti-PKN1 (Becton Dickinson, BD 610687; 1:1,000 dilution). Cell lysates were reprobed with a mouse monoclonal antibody anti-β-actin (clone C4, Millipore; 1:5,000 dilution). Full Western blot images are presented in Source Data.

**Cytokine detection and cell death assay**

Levels of IL-1β in monocyte supernatant were quantified by ELISA (R&D Systems). IL-18 levels were quantified using the following

antibodies for capture and detection: anti-human IL-18 antibody (4 μg/ml; # D044-3, MBL, Woburn, MA, USA) and anti-human IL-18 antibody coupled to biotin (20 ng/ml, # D045-6, MBL). Cell death was monitored by incubating $2 \times 10^4$ monocytes per well of a black 96-well plate (Costar, Corning) with propidium iodide (PI, Sigma) at 5 μg/ml. Three technical replicates per conditions were done. UCN-01 was added at 12.5 μM in the absence of any priming signal. Nigericin was added at 5 μM after a 3-h priming with LPS at 10 ng/ml. Real-time PI incorporation was measured every 5 min from 15 min to 105 min (for primary monocytes) or from 0 to 180 min (for U937 cells) post-nigericin/UCN-01 intoxication on a fluorimeter (Tecan) using the following wavelengths: excitation 535 nm (bandwidth 15 nm) and emission 635 nm (bandwidth 15 nm) (Case & Roy, 2011; Pierini et al, 2012). Cell death was normalized using PI incorporation in monocytes treated with Triton X-100 for 15 min (=100% cell death). Of note, PI value can artificially decline after several hours. As a further correction, the first time point of the kinetics was set to 0. When the reading was stopped (for < 5 min) to add UCN-01 at 20 h post-doxycycline addition, the values after UCN-01 addition were corrected so that the values right after UCN-01 addition matched the values obtained right before. The areas under the curve were computed using the trapezoid rule (Prism 6; GraphPad). Kinetics until 60 and 105 min post-UCN-01 and post-nigericin treatments, respectively, were retained. To extract the time corresponding to 20% cell death, a non-linear regression analysis (Prism 6; GraphPad) was used to fit a sixth-order polynomial curve to the normalized cell death kinetics using the least squares fit as the fitting method. The obtained curve was used to interpolate the time corresponding to 20% cell death.

## Statistics

For HD/patient data, normal distribution was verified using D'Agostino–Pearson omnibus normality test. When Gaussian distribution could not be demonstrated for the compared variables, non-parametric tests were used. Results were expressed as median ± interquartile range. When two groups (e.g., HD and FMF), including at least one that did not pass the normality test, were compared, Wilcoxon rank-sum test was performed, and two-tailed p-values are shown. When two groups (e.g., HD and FMF) that passed the normality test were compared, unpaired t-test was performed, and two-tailed $P$-values are shown. When multiple comparisons between groups (e.g., HD and FMF) that all passed the normality test were made, one-way ANOVA with Sidak's multiple comparison tests was performed. When multiple comparisons between groups (e.g., HD and FMF), some of them that did not pass the normality test, were made, Kruskal–Wallis with Dunn's multiple comparison tests were performed. Multiplicity-adjusted $P$-values are shown. When multiple comparisons between paired groups of identical numbers (e.g., FMF and FMF + inhib.), some of them that did not pass the normality test, were made, Friedman with Dunn's multiple comparison tests were performed. Multiplicity-adjusted $P$-values are shown. When a comparison between two paired groups (e.g., FMF and FMF + inhib.), at least one of each did not pass the normality test, was made, Wilcoxon matched-pairs signed rank test was used. Two-tailed $P$-value is shown. Correlation significance was determined by Spearman test. The predictive power of the mechanisms of interest was assessed using receiver operating curve (ROC) analyses. The area under the ROC (AUC) and its 95% CI were estimated for each

### The paper explained

#### Problem

Familial Mediterranean fever (FMF) is the most frequent hereditary systemic autoinflammatory disease. FMF is, in most cases, associated with biallelic mutations of the *MEFV* gene encoding Pyrin, an inflammasome sensor. The link between *MEFV* mutations and the dysregulated Pyrin inflammasome activation observed in FMF patients is unclear. Furthermore, most of the 365 described *MEFV* variants are of uncertain significance and the genetic validation of the clinical FMF diagnosis remains challenging.

#### Results

Thanks to the use of protein kinase inhibitors and phospho-null mutants; this study demonstrates that Pyrin dephosphorylation triggers full inflammasome activation in FMF patients' monocytes but not in healthy donors' monocytes. Furthermore, the pathogenic p.M694V *MEFV* mutation, most frequently observed in FMF patient, triggers constitutive inflammasome activation only when combined to phospho-null *MEFV* mutations. These results indicate that Pyrin inflammasome activation is controlled by two independent mechanisms in healthy donors but only by phosphorylation/dephosphorylation in FMF patients. This difference can be exploited using protein kinase inhibitors and primary monocytes from patients to functionally diagnose FMF.

#### Clinical impact

This study increases our knowledge of the molecular mechanisms underlying inflammation in FMF patients. Furthermore, it should improve FMF diagnosis by providing a quick and easy test in primary monocytes to discriminate FMF patients from patients with unrelated inflammatory conditions. Finally, this study provides means to determine the pathogenicity of *MEFV* variants, which should reduce in the future the number of variants of uncertain significance and improve genetic testing.

mechanism. Threshold values for the prediction of patients with FMF were determined by maximizing the Youden index. All tests were two-sided at the 0.05 significance level. Correlation coefficient was computed using non-parametric Spearman correlation. Statistical analyses were carried out with Prism 6 and R version 3.1.1 (http://www.R-project.org).

## Study approval

The study was approved by the French Comité de Protection des Personnes (CPP,#L16-189) and by the French Comité Consultatif sur le Traitement de l'Information en matière de Recherche dans le domaine de la Santé (CCTIRS, #16.864). The experiments conformed to the principles set out in the WMA Declaration of Helsinki and the Department of Health and Human Services' Belmont Report. HD blood was provided by the Etablissement Français du Sang in the framework of the convention #14-1820. Informed consent was received from participants prior to inclusion in the study.

# Data availability

All data are available within the manuscript, its appendix, and Source Data files.

**Expanded View** for this article is available online.

## Acknowledgements

This work was performed in the framework of the Centre National de Reférence RAISE. We acknowledge the contribution of the Etablissement Français du Sang Auvergne—Rhône-Alpes and of SFR Biosciences (UMS3444/CNRS, US8/Inserm, ENS de Lyon, UCBL) Cytometry, Microscopy, and Vectorology facilities. Funding: This work is supported by an ANR grant (FMFgeneToDiag #ANR-17-CE17-0021), an ERC-2012-StG_3115 and funding from the European Union's Horizon 2020 research and innovation program under grant agreement #779295 (Immu-nAID).

## Author contributions

FM, LL, SB, TM, LW, AM, and DC performed the experiments; SK performed the statistical analysis. MD, AD, MG-V, AL, and PS provided substantial clinical inputs on the work; M-RP provided key reagents; TW, AB, YJ, and TH designed and interpreted the work, YJ and TH wrote the manuscript, all authors reviewed and approved the manuscript.

## Conflict of interest

FM, LL, AM, AB, YJ, and TH are listed on a patent related to FMF diagnosis.

## For more information

(i) FMF description on Orphanet (the portal for rare diseases and orphan drugs) https://www.orpha.net/consor/www/cgi-bin/OC_Exp.php?lng=EN&Expert=342

(ii) FMF OMIM (An Online Catalog of Human Genes and Genetic Disorders): https://omim.org/entry/249100

(iii) Patient association: FMF and AID Global association: https://www.fmfandaid.org/

(iv) *MEFV* sequence variants (Infevers, the registry of Hereditary Auto-Inflammatory Disorders Mutations) https://infevers.umai-montpellier.fr/web/search.php?n=1

(v) CIRI website: http://ciri.inserm.fr/en/

(vi) ImmunAID website: Immunome project consortium for AutoInflammatory Disorders https://www.immunaid.eu/

(vii) International Patent filed from this work. https://patentscope.wipo.int/search/en/detail.jsf?docId=WO2019048569

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
