## [Review Process File · EMBO Molecular Medicine]

Pyrin dephosphorylation is sufficient to trigger inflammasome activation in Familial Mediterranean Fever patients

Flora Magnotti, Lucie Lefevre, Sarah Benezech, Tiphaine Malsot, Louis Waeckel, Amandine Martin, Sébastien Kerever, Daria Chirita, Marine Desjonqueres, Agnès Duquesne, Mathieu Gerfaud-Valentin, Audrey Laurent, Pascal Sève, Michel-Robert Popoff, Thierry Walzer, Alexandre Belot, Yvan Jamilloux, Thomas Henry

Review timeline:

Submission date:	6 March 2019
Editorial Decision:	15 April 2019
Revision received:	24 July 2019
Editorial Decision:	22 August 2019
Revision received:	4 September 2019
Accepted:	13 September 2019

Editor: Lise Roth

Transaction Report:

1st Editorial Decision

15 April 2019

Thank you for the submission of your manuscript to EMBO Molecular Medicine. We have now heard back from the referees whom we asked to evaluate your manuscript.

As you will see from the reports below, the 3 referees acknowledge the potential interest of the findings for the field, however they also have fundamental concerns that should be addressed in a major round of revision of the present manuscript. In particular, the molecular mechanism needs to be convincingly strengthened to fully support the conclusions.

Addressing the reviewers' concerns in full will be necessary for further considering the manuscript in our journal. EMBO Molecular Medicine encourages a single round of revision only and therefore, acceptance or rejection of the manuscript will depend on the completeness of your responses included in the next, final version of the manuscript.

Please also contact us as soon as possible if similar work is published elsewhere. If other work is published, we may not be able to extend the revision period beyond three months.

I look forward to receiving your revised manuscript.

***** Reviewer's comments *****

Referee #1 (Remarks for Author):

Magnotti et al. provide compelling evidence for a dual inhibition mechanism of Pyrin where dephosphorylation upon PKC inhibition is only sufficient for activation in the presence of patient mutations affecting the second mechanisms. This could serve as a diagnostic tool to identify and stratify FMF patients. The experiments are well performed and support the conclusion. The manuscript is well written and convincing. We have only minor comments:

Could a dose dependency of the inhibitor be involved in the differences observed between FMF and HD? The authors should more systematically titrate their inhibitors and perform dose-response and time-course experiments to check whether HD cells will secrete substantial IL-1 under such conditions (similar to SID, but going higher and including several time-points and other PKC inhibitors). Similar experiments are contained in F1D, G for cell death. What is the maximum cell death rate reached in HD cells when using this read-out? How do these results look if LDH release (late) is used as a readout instead of PI uptake (early)?

The main text provides very little explanation on F2A-C, please expand. Please consider that the FLICA assay is not specific for Casp1 but also yields a signal upon casp8 activity, which might be important when making conclusion concerning other forms of cell death involved.

Please make sure that the figure legends mention the method used for each panel (e.g. Fig 2D (FACS?). Please provide the original data for Fig. 2D, not just bar graphs.

The legends are missing the black and red boxes in F3A, B

Sentence "Surprisingly, colchicine strongly reduced (86%) both IL-1 β release and fully abolished pyroptosis upon PKC inhibition (Fig. 3B-E)." - maybe expand a little bit at this point, why (in contrast to what) this is surprising to increase accessibility for non-inflammasome experts. Fig 3B seems to be the central panel of the study and the reader should get it.

F4A-C, if these are separate panels, the Y-axis label should be included in each of them. In panel B, the "i" in Nig is missing.

The label of F4H is unclear, although one can guess that it refers to the neighboring panel. We would suggest to make stand-alone labelling for panels with separate call-outs.

Please check the use of commas (or the lack thereof), e.g. before "suggesting"

Sentence "Importantly, UCN-01 had no cytotoxic action on the p.S242R- or the p.S242R/p.M694V-expressing cell lines,..." maybe expand a bit more: the p.S242R cells do not die in the presence of UCN-01, while p.S242R/p.M694V are already dead before the compound is added - and do not "die more" afterwards, is what is probably meant here.

Please check the use of the plural of inhibitor, e.g. in "PKC inhibitors response functionally discriminates FMF patients from patients with unrelated inflammatory conditions", or the last sentence on page 12.

F5A-E, the meaning of the label "DC" should be mentioned in the main text on page 12 to increase accessibility.

The story is straight-forward and convincing and the discussion might be shortened.

Referee #2 (Comments on Novelty/Model System for Author):

The authors have not indicated why they performed statistics (non-parametric test) the way they did. Also, for the kind of statistics they have performed, summary data should be shown as median values than means. Matched data (with symbols with different colours/shapes indicating a given

patient) should be shown for better clarity for the informed reader. Cell line systems and other assays are perfect for the questions being addressed.

Referee #2 (Remarks for Author):

In their manuscript the authors present data on FMF patient-derived cells and the selective early activation of pyroptotic cell-death in these cells in response to relatively broad-spectrum PKC-inhibitors. They try 2 different inhibitors and measure pyroptosis, IL-1b release, ASC specks and caspase-1 activation in cells from patients and healthy-control. They show that in cells reconstituted with M694V variant, PKC inhibitors are sufficient to activate the Pylrin inflammasome. In addition, this can be mimicked by phospho-incompetent mutation at S242 in the M694V allele. This part of the manuscript is largely supported by data presented. However, there is little mechanistic insight on how M694V mutation uncouples activation from microtubule-dynamics. Also lacking are data with siRNA/inducible-shRNA approaches, when CRISPR approaches failed. Furthermore, only some of the patients used have a homozygous M694V allele, and whether those with biallelic/combinational mutations are also activated similarly is not formally demonstrated.

Major comments:

1. Some of the data presented are similar to those reported previously in mouse or human systems and distracts from the main message of this work. In particular, data on FMF mutations have identified the activation of the Pylrin inflammasome in response to similar inhibitors (STS) or activators (bryostatin) (Park et al). The inhibition of Pylrin inflammasomes by colchicine is also well known. The hyper-activation of the Pylrin inflammasome in cells from patients is also not surprising and has been shown by other groups (e.g. Van Gorp et al; cited). Much of the focus here is/should be on phosphorylation, but this needs to be developed further as the data on reconstitution of with the M694V mutation, while new and interesting, are not sufficiently detailed to provide significant new insight.
2. The authors should tone-down their claims of a diagnostic test based on these results.
3. It is not clear which data points are from which patients/genotypes, and within data panels, which cells showed certain levels of IL-1b, cell death or TNF or other parameters they measured. Data could be better depicted with symbols with matched colour and/or shapes so interested and informed readers may see whether cells that undergo most pyroptosis, also release most TNF/IL-1b. It is most important to indicate which genotypes were used in which experiments across figures, especially if the authors wish to make generalised statements on diagnostics (which I don't think they should).
4. The authors do not describe why non-parametric tests were used. Did they check data distribution and data-transforms to then use parametric tests if possible? Even so, as they have performed non-parametric tests, they must show medians and quartile ranges and not means/SEM. Also, Some data panels lack legends e.g. black and red in Fig 3A-B.
5. Authors should develop the molecular mechanistic side of their manuscript. For example, is Pylrin dephosphorylated in patient-derived cells upon PKC-inhibitor treatment (Phos-tag gels or their phospho-antibody?) - this has not been formally demonstrated in this work. How do homozygous and/or mixed mutations affect this? Have the authors attempted to reconstitute MEFV^{-/-} cells with other mutations or mixed mutations (e.g. express two variants with a P2A self-cleavage tag) that match their patient samples - this would be needed if they wish to draw broad conclusions that PKC inhibitors can distinguish FMF patients.
6. As some of their data conflict previous data on colchicine efficacy, they should address this as they have access to patient-derived cells and good molecular cell-line systems and requisite tools. Have they checked other inhibitors, concentrations curves, silencing of PKN1/2? There is evidence that the mouse Mefv^{V726} mutants interact with PKN1 more after colchicine treatment. This would indicate that colchicine is not necessary acting at a second step, but increases inhibitory phosphorylation of Pylrin. Mechanistically, does the M694V/S242A not interact with 14-3-3 proteins (this has been suggested before Park et al)? Does phosphorylation status/interaction with 14-3-3 of Pylrin/M694V etc. change after colchicine treatment - is that the reason colchicine works in some cases and not others? Such experiments will provide more insight on this very interesting molecule.

Referee #3 (Remarks for Author):

In the manuscript by Magnotti and colleagues a novel mechanism explaining pyrin hyperactivation in patients with FMF is proposed. The authors postulate that FMF-mutant Pyrin can be activated solely by dephosphorylation at position Ser242. They demonstrate by using small molecule inhibitors of PKC superfamily kinases that FMF mutant pyrin can be activated, yet the same inhibitors (under the conditions chosen) fail to activate wild-type pyrin. This data suggests that in FMF patients one of two safeguard mechanisms, that normally prevents pyrin activation, is missing.

The paper is of potential interest to the community as it brings new mechanistic insights into a fairly common genetic disease. Very few papers have provided a similarly deep analysis of human pyrin activation in primary patient-derived cells. While the work is of interest a number of issues require attention. Most critically a clear demonstration that the proposed mechanism is indeed correct is missing.

Major points:

- The authors make the claim that pyrin dephosphorylation of S242 in FMF mutant pyrin is the mechanism by which PKC inhibitors can activate mutant pyrin. A clear biochemical characterization that these inhibitors act directly by inhibiting pyrin dephosphorylation needs to be performed. The WB shown is not very convincing. Addition of UCN-01 already reduces the amount of pyrin. What is the basis for that? It is also not clear which of the multiple bands that show up in the WB is in fact pyrin. The authors should generate a non-phosphorylatable form of pyrin and test the function and the validity of the pPyrin Ab.
- It is interesting that colchicine can block PKC induced FMF Pyrin activation. What is the mechanism? Can the authors test whether colchicine interferes with pyrin dephosphorylation?
- The spread of "responders" to the PKC compound for the FMF patients is very large and there are clearly some that don't respond to the treatment at all. I think this demonstrates that at least for some donors both, the proposed mechanism is incorrect as well as that the advocated test will not be as conclusive as proposed by the authors.
- PKC inhibitors can promote noncanonical caspase-8 dependent release of IL-1 β . Can the authors exclude the reported inflammasome independent non-canonical processing of IL-1 β ?

Minor points:

- The data representation is inconsistent. Please select one representation style. It would be good if for the inhibitor experiments would make use of unique symbol or colors for each of the donors to know which comparison to make.
- Genetic evidence for the proposed mechanism is missing. The authors report that the attempts to knock-out PKN1/2 failed. Yet, there are alternative ways to generate a transient knock down. This work is essential, especially as the inhibitors used have many effects.

Specific for figures:

Figure 2:

- Please include all staining controls for the ASC specks? This could at least be added to the supp figures. A quantitation of speck like objects in the secondary alone control cells would be appreciated.
- Inhibition using VX-765 and ZVAD would be useful to further define the type of cell death and back up their current data.

Figure 3:

- Controls are needed for the effect of colchicine.

Figure 4:

- Vector alone control is lacking for all experiments
- Figure 4D, how do the M694V + Dox cells secrete IL-1 in the absence of an LPS priming signal (and quite a lot, 500 pg/ml)?
- Figure 4G - Do the authors change the scale of the graph for the UCN-01 addition? At this point eh p.S242R seems to change position relative to the WT (shows lower values rather than higher).
- Figure 4 H would be clearer if they labeled the legend fully. Furthermore, Wilcoxon rank-sum test (Mann-Whitney U test) requires that the samples are non-related. For related or matched samples (in a not normally-distributed population), a Wilcoxon signed-rank test is more convenient. This is the case when comparing Dox and Dox+UNC-01 conditions in the same sample.

1st Revision - authors' response

24 July 2019

Referee #1 (Remarks for Author):

Magnotti et al. provide compelling evidence for a dual inhibition mechanism of P2Y₆ where dephosphorylation upon PKC inhibition is only sufficient for activation in the presence of patient mutations affecting the second mechanisms. This could serve as a diagnostic tool to identify and stratify FMF patients. The experiments are well performed and support the conclusion. The manuscript is well written and convincing.

We thank the reviewer for his positive evaluation of our work.

We have only minor comments:

1-Could a dose dependency of the inhibitor be involved in the differences observed between FMF and HD? The authors should more systematically titrate their inhibitors and perform dose-response and time-course experiments to check whether HD cells will secrete substantial IL-1 under such conditions (similar to SID, but going higher and including several time-points and other PKC inhibitors). Similar experiments are contained in F1D, G for cell death. What is the maximum cell death rate reached in HD cells when using this read-out? How do these results look if LDH release (late) is used as a readout instead of PI uptake (early)?

We now provide in Expanded view EV1A a dose and time-response with staurosporine. For both staurosporine and UCN-01, we show that the difference of IL-1 β response is still present at 10-fold the concentration used in our main manuscript. We cannot go higher in terms of concentration due to the DMSO solvent going above 1% final.

HD monocytes cell death can reach 100% using PI incorporation (likely due to the late membrane permeabilization observed during apoptosis-see Fig 2), although we observed a large inter-individual variation. Individual curves now presented in Appendix Fig S1D should clarify this.

We did not extensively use LDH release assay since it requires at least 5×10^4 cells compared to 10^4 and 2×10^4 for IL-1 β ELISA and RT cell death assay (PI incorporation), respectively. Cell death based on LDH assay was calculated as follows:

Staurosporine 1.25 μ M, 3 h of treatment, one HD: 0.61 \pm 4.5 %;

UCN-01 12.5 μ M, 3 h of treatment, one HD: 5.34 \pm 0.97 %;

These results are consistent with the lack of early (1.5 h) cell death noted in HD monocytes using PI incorporation.

Altogether, these results show that even at a 10-fold higher concentration of staurosporine or UCN-01 (12.5 μ M staurosporine or 125 μ M UCN-01) there is still a large difference between the HD's and the FMF patients' monocytes response. Similarly, collecting supernatant at 3 h post-treatment instead of 90 minutes does not affect the difference between HD's and the FMF patients' monocytes response, strongly suggesting that the difference is not due to a difference in the dose response.

2-The main text provides very little explanation on F2A-C, please expand.

We have expanded this part both in the main section and in the method section.

3-Please consider that the FLICA assay is not specific for Casp1 but also yields a signal upon casp8 activity, which might be important when making conclusion concerning other forms of cell death involved.

We do agree that most caspase inhibitors and caspase fluorescent probes are poorly specific since they rely on a short recognition peptide (e.g. YVAD in the case of FLICA caspase-1). We have previously reported such non-specific staining in Pierini *et al.* (Pierini *et al.*, 2012). Yet, we are confident that the phenotype we describe here is caspase-1/Gasdermin D dependent since invalidation of caspase-1 or Gasdermin D in U937 cells fully abolishes the early cell death and IL-1 β release (Fig. 4C).

While we can not exclude a concomitant activation of caspase-8 (as previously reported by us and others (Pierini *et al.*, 2012; Man *et al.*, 2014; Sagulenko *et al.*, 2013), we strongly believe the cell death phenotype is mostly driven by caspase-1 based on the absence of apoptotic cells (Annexin-V+ PI neg cells) upon UCN-01 treatment in FMF monocytes.

Fig EV1G of the revised manuscript presents the effect of Casp1, 3 and 8 inhibitors on IL-1 β release. Only caspase-1 inhibitor demonstrated a robust effect at low concentrations (5 μ M), strongly suggesting that the response observed in primary FMF patients' monocytes is mediated by caspase-1 and not by caspase-8. We could not test caspase-8 inhibition in regards to cell death due to the triggering of necroptosis upon caspase-8 inhibition (Kaiser *et al.*, 2011)(see response to reviewer 3).

We have expanded this part in the manuscript to make the reader aware of the nature of the fluorescent probe *"The UCN-01-mediated induction of inflammasome activation (...) was (...) confirmed by quantifying (...) active caspase-1 using the fluorescent inhibitor probe FAM-YVAD-FMK, referred to as FLICA-Casp1"*

4-Please make sure that the figure legends mention the method used for each panel (e.g. Fig 2D (FACS?). Please provide the original data for Fig. 2D, not just bar graphs.

We have mentioned in the revised manuscript the method used for each panel (including Flow cytometry for Fig 2D). The original FACS plot are now shown in the revised Fig. 2D and the gating strategy in Data Source Fig 2D.

5-The legends are missing the black and red boxes in F3A, B

Legends have been added in all the panels

6-Sentence "Surprisingly, colchicine strongly reduced (86%) both IL-1 β release and fully abolished pyroptosis upon PKC inhibition (Fig. 3B-E)." - maybe expand a little bit at this point, why (in contrast to what) this is surprising to increase accessibility for non-inflammasome experts. Fig 3B seems to be the central panel of the study and the reader should get it.

We have expanded this part to better explain the context and take into account the referee's comment. The revised section reads as follows:

*"Colchicine specifically blocks the Pypin inflammasome in murine macrophages and in PBMCs from healthy donors (Van Gorp *et al.*, 2016; Park *et al.*, 2016; Gao *et al.*, 2016). Despite its long-demonstrated clinical efficacy in FMF patients, colchicine was recently shown to be inefficient to block IL-1 β release in PBMCs from FMF patients exposed to TcdA (Van Gorp *et al.*, 2016). Although we did see a partial (40%) inhibition of TcdB-mediated IL-1 β response in monocytes from FMF patients, this inhibition was almost total (86%) in HD monocytes, thus confirming that toxin-mediated Pypin inflammasome activation is less sensitive to colchicine inhibition in FMF patients than in HD monocytes (Fig. 3B). In contrast to the lack of (Van Gorp *et al.*, 2016)/partial (our result) inhibition observed upon TcdA/B-mediated Pypin inflammasome activation, colchicine strongly reduced (86%) both IL-1 β release and fully abolished pyroptosis upon PKC inhibition (Fig. 3B-E)."*

7-F4A-C, if these are separate panels, the Y-axis label should be included in each of them. In panel B, the "i" in Nig is missing.

The Y-axis label is included in each panel of the revised manuscript. The "i" in Nig has been added in all figure panels.

8-The label of F4H is unclear, although one can guess that it refers to the neighboring panel. We would suggest to make stand-alone labelling for panels with separate call-outs.

Each panel in the revised manuscript has its own legend.

9-Please check the use of commas (or the lack thereof), e.g. before "suggesting"

Commas have been added before participial phrases, including the ones starting with "suggesting".

10-Sentence " *Importantly, UCN-01 had no cytotoxic action on the p.S242R- or the p.S242R/p.M694V-expressing cell lines,...*" maybe expand a bit more: *the p.S242R cells do not die in the presence of UCN-01, while p.S242R/p.M694V are already dead before the compound is added - and do not "die more" afterwards, is what is probably meant here.*

This section has been extensively modified in the revised manuscript. The specific sentence referring to your point reads as follows in the revised version:

"No additional cytotoxic effect could be detected on the p.S208C/S242R/M694V-expressing cell line since all the cells were dead at 20 h post-doxycycline addition"

11-Please check the use of the plural of inhibitor, e.g. in *"PKC inhibitors response functionally discriminates FMF patients from patients with unrelated inflammatory conditions"*, or the last sentence on page 12.

The revised manuscript has been edited and now reads:

"PKC inhibitor responses functionally discriminate (...)"
"none of the non-FMF patients (...) responded to the two PKC inhibitors by (...)" 12-F5A-E, the meaning of the label "DC" should be mentioned in the main text on page 12 to increase accessibility.

Done

13-The story is straight-forward and convincing and the discussion might be shortened.

We would rather not shorten the discussion which reflects comments from colleagues or from previous reviewers, but would be willing to consider editorial request to move some of the discussion in the Appendix.

Referee #2 (Comments on Novelty/Model System for Author):

1-The authors have not indicated why they performed statistics (non-parametric test) the way they did. Also, for the kind of statistics they have performed, summary data should be shown as median values than means.

We have developed the section on statistical analysis in the revised manuscript (Methods section and each figure legend) and present graphs with median values +/- interquartile range whenever normal distribution could not be demonstrated (see below for an extended response).

2-Matched data (with symbols with different colours/shapes indicating a given patient) should be shown for better clarity for the informed reader.

Matched data with symbols with different shapes indicating a given patient are shown in the revised manuscript for Fig. 2B, 2C, 3, the latter which includes treated and untreated samples. Due to

visibility/clarity issues, this was not possible for Fig 1. We have thus decided to present in Appendix Figure S1 and S4, enlarged figures with each patient (genotype indicated) individualized. Further, when the number of patients allowed it, we have included, in the figure legend (section data information), the patient number and her/his genotype.

3-Cell line systems and other assays are perfect for the questions being addressed.

Thank you!

Referee #2 (Remarks for Author):

1-In their manuscript the authors present data on FMF patient-derived cells and the selective early activation of pyroptotic cell-death in these cells in response to relatively broad-spectrum PKC-inhibitors. They try 2 different inhibitors and measure pyroptosis, IL-1b release, ASC specks and caspase-1 activation in cells from patients and healthy-control. They show that in cells reconstituted with M694V variant, PKC inhibitors are sufficient to activate the Pyrin inflammasome. In addition, this can be mimicked by phospho-incompetent mutation at S242 in the M694V allele. This part of the manuscript is largely supported by data presented.

We are glad to read that the reviewer is convinced that the cell line systems, chemical and genetic evidence supports our conclusion that Pyrin dephosphorylation is sufficient to promote inflammasome activation in monocytes from FMF patients. We do believe this message is very important since it explains the hyper-reactivity of the Pyrin inflammasome in this auto-inflammatory disease.

2-However, there is little mechanistic insight on how M694V mutation uncouples activation from microtubule-dynamics.

We do agree with the referee that how M694V mutation uncouples activation from microtubule-dynamics is still fully unclear and is a fantastic challenge for the field to understand. We believe this is out of the scope of the current manuscript which is focusing on the impact of Pyrin dephosphorylation on inflammasome activation.

3-Also lacking are data with siRNA/inducible-shRNA approaches, when CRISPR approaches failed.

We now provide data in the revised manuscript (Figure EV5) in which we performed *PKN2* siRNA experiments in a *PKN1*KO cell line stably expressing either WT Pyrin or p.M694V Pyrin. Although cell death levels are relatively low (reaching 20%), the increased cell death is specific to i) the presence of Doxycycline ii) the p.M694V mutation iii) an efficient (and *PKN2*-targeting) siRNA. Although we were cautious in the interpretation of this challenging experiment, these results strengthen our genetic approach with the phospho-null mutations.

4-Furthermore, only some of the patients used have a homozygous M694V allele, and whether those with biallelic/combinational mutations are also activated similarly is not formally demonstrated.

As of today, 349 *MEFV* variants are listed in Infevers, the registry of hereditary autoinflammatory disorders mutations (<https://infevers.umai-montpellier.fr/web/search.php?n=1>). Testing each mutations (not to say the combination of bi-allelic mutations or complex alleles) is not feasible at this stage.

We do agree this will be important to perform in the future, and our strategy is first to publish the current demonstration with a limited number of FMF patients (n=39) of various genotypes (18 of them being homozygous for the p.M694V allele) in order to perform a multicentric recruitment in different countries to assess the genotype/phenotype correlation.

Of note, since our cohort contains 21 FMF patients harbouring other mutations than the biallelic p.M694V mutations, we strongly believe the response described here extends to other genotypes. The analysis below (corresponding to Figure 1A) is performed on the 19 analysed FMF patients that do not possess the biallelic p.M694V mutation and demonstrates that our conclusions likely extend beyond the bi-allelic p.M694V mutation.

Staurosporine triggers IL-1 β secretion in FMF patients not bearing the bi-allelic p.M694V mutation (related to Figure 1A of the revised manuscript).

FMF patients not bearing the bi-allelic p.M694V mutation were included in the comparison with the same criteria as Fig 1A. The values did not pass the D'Agostino and Pearson omnibus normality test and were compared using Wilcoxon rank-sum test (two-tailed p-value is shown, ***p<0.001)

In the revised manuscript, we have added in the appendix Figure S1D, the individual assays for each patient and their "same day" healthy donors. The limit of the conclusions for each individual genotypes is discussed in the manuscript.

“Although our data suggest that a single clearly pathogenic MEFV variant is sufficient to confer to PKC inhibitors the ability to trigger inflammasome activation, our cohort of FMF patients is currently too small to draw robust conclusions regarding the number (biallelic mutations vs. mono-allelic) of clearly pathogenic variants. Furthermore, due to the large number of MEFV variants present in our cohort, conclusions on specific MEFV variants will require the recruitment of a large number of FMF patients coupled to the functional characterization of MEFV variants in genetically-engineered cell lines.”

Finally, in an effort to further validate our findings with different mutations, we have generated cell lines expressing two other clearly pathogenic MEFV variants p.M694I and p.M680I and a variant of unknown significance p.P369S. We demonstrate that the response to UCN-01 is similar for the cell lines expressing the two clearly pathogenic variants to the one observed in the p.M694V-expressing cell line. In contrast, we did not detect such a response in the p.P369S-expressing cell line indicating either an absence of pathogenicity or a different level/mechanism of pathogenicity. This new piece of data is presented in Figure EV4. The revised manuscript now reads:

“Similar results were obtained with the two clearly pathogenic MEFV variant p.M694I and p.M680I (Fig. EV4A-H). In contrast, expression of the variant of unknown significance p.P369S did not trigger the same response suggesting that it is either a non-pathogenic variant (in line with its higher frequency in the human population Fig. EV4I), that its pathogenicity is undetectable in our experimental system or that its pathogenicity is due to another molecular mechanism.”

Major comments:

1A. Some of the data presented are similar to those reported previously in mouse or human systems and distracts from the main message of this work. In particular, data on FMF mutations have identified the activation of the Pyrin inflammasome in response to similar inhibitors (STS) or activators (bryostatin) (Park et al).

We do not use bryostatin in the current work We do (and did) acknowledge the work by Park and colleagues and how our study built on the seminal results described in 2016 in Nature Immunology (the first sentence of the result section reads as follows:

“The current model for Pypin inflammasome activation indicates that activation results from the dephosphorylation of Pypin following the lack of sustained activation of PKN1/2, two kinases from the PKC superfamily (Park et al, 2016).”

Park and colleagues have tested staurosporine in WT murine BMDM (fig 3A) and have demonstrated that staurosporine activates the Pypin inflammasome in WT macrophages (that harbours a very different *MEFV* gene). To my knowledge, Park and colleagues have shown that PKN activators (Bryostatin) can decrease constitutive IL-1 β release in FMF patients (Fig 4D, E) but they did not test PKC inhibitors in FMF patients.

Our results are novel in comparison from the work from Park and colleagues since we demonstrate that PKC superfamily inhibitors do not activate substantially the Pypin inflammasome in healthy donor monocyte while they fully activate the Pypin inflammasome in FMF patients monocytes.

1B-The inhibition of Pypin inflammasomes by colchicine is also well known.

The inhibition of the Pypin inflammasome by colchicine in mouse macrophages and healthy donor monocytes/PBMC is indeed well known.

Yet, a recent PNAS paper by van Gorp and colleagues demonstrated that colchicine can not inhibit TcdA-mediated Pypin inflammasome in PBMCs from FMF patients ("*FMF mutations render Pypin activation resistant to colchicine blockade*" is one of the conclusion of the van Gorp et al. paper). We do believe demonstrating that colchicine is effective in blocking the Pypin inflammasome in FMF patients' monocytes is very important to reconcile the clinics and the experimental work and to illustrate the complexity of Pypin inflammasome activation.

1C-The hyper-activation of the Pypin inflammasome in cells from patients is also not surprising and has been shown by other groups (e.g. Van Gorp et al; cited).

We have to disagree with the referee. Van Gorp et al. have not seen any hyper-activation of the Pypin inflammasome in cells from patients: see below for sentences from the Van Gorp et al. PNAS paper (Van Gorp et al, 2016):

*“The *C. difficile* infection triggered a substantial but **comparable** release of IL-1 β in wild-type and FMF PBMCs (Fig. 2A). FMF PBMCs that had been intoxicated with TcdA also secreted **normal** levels of IL-1 β (...) (Fig. 2B). Moreover, IL-1 β levels secreted by FMF (...) were **comparable to those of healthy donors** (Fig. 2C). These results indicate that FMF mutations are **not hypermorphic for inflammasome activation** relayed by either Pypin or NLRP3”*

At high concentration of *C. difficile* toxins, we have also observed a normal response of the Pypin inflammasome as described in Van Gorp et al. Yet, we have reported that the Pypin inflammasome has a decrease threshold of activation when stimulated with one natural stimulus TcdA (Jamilloux et al, 2017). This paper does not contain mechanistic insights. Hyper-activation of the NLRP3 inflammasome (not of the Pypin inflammasome) in FMF patients have been reported by Omenetti and colleagues (Omenetti et al, 2014).

So we do believe our work is novel and important in demonstrating the hyper-activation of the Pypin inflammasome in response specifically to PKC-superfamily inhibitor.

1D-Much of the focus here is/should be on phosphorylation, but this needs to be developed further as the data on reconstitution of with the M694V mutation, while new and interesting, are not sufficiently detailed to provide significant new insight.

First I would like to emphasize that we had a substantial amount of mechanism in the previous manuscript: Except for Fig.1 and 5 that described the phenotype in primary cells from patients in comparison with healthy donors and patients with other inflammatory conditions, Fig. 2, 3 and 4 were all about mechanistic and provided the following insights.

1. Position in regards to ASC speck formation (an important feature to investigate in primary human monocytes (Gaidt et al, 2016))

2. Sensitivity to colchicine (an important feature to investigate due to controversy (Van Gorp *et al.*, 2016))
3. Phenotype fully recapitulated in a genetically tractable system by expression of p.M694V Pyrin
4. Cell death fully dependent on both GSDMD and Casp1 (a perhaps expected dependency but important to demonstrate since UCN-01/Staurosporine can trigger apoptosis)
5. Similar dephosphorylation upon UCN-01 treatment in WT and p.M694V-expressing Pyrin
6. Genetic proof of the mechanism of the inhibitor using the phospho-null mutant p.S242R in combination or not with p.M694V mutation.
7. Demonstration that colchicine acts independently of the phosphorylation site p.S242R

In the revised manuscript, we have strengthened this mechanistic part by:

1. adding a control of the specificity of the phospho-antibody
2. extending the mechanism to p.M694I and p.M680I Pyrin variants
3. providing evidence that PKN1/2 inhibition mediates the UCN-01-mediated phenotype
4. demonstrating the synergy between S208 and S242 phospho-null mutations in combination with p.M694V mutation to trigger cell death and IL-18 production.

A novel figure (Figure 5) in the revised manuscript presents these new findings. We believe the revised manuscript provides significant insights into the cellular and molecular consequences of some of the most pathogenic *MEFV* mutations.

2. The authors should tone-down their claims of a diagnostic test based on these results.

We have modified the text accordingly in several places including by adding the following section on the limit of the current study. We agree with the reviewer that this needs to be thoroughly tested in the future.

*“Of note, several limitations of the current study remain to be overcome to fully evaluate the diagnostic potential of this assay. First, the sensitivity of the test has to be defined in respect to the different *MEFV* genotypes. Second, the assay would have to be validated in whole blood to be compatible with clinical laboratory. Third, the test remains to be validated in a large independent, multicentric cohort of patients in a prospective manner.”*

3. It is not clear which data points are from which patients/genotypes, and within data panels, which cells showed certain levels of IL-1b, cell death or TNF or other parameters they measured. Data could be better depicted with symbols with matched colour and/or shapes so interested and informed readers may see whether cells that undergo most pyroptosis, also release most TNF/IL-1b. It is most important to indicate which genotypes were used in which experiments across figures, especially if the authors wish to make generalised statements on diagnostics (which I don't think they should).

In Figure 3 of the revised manuscript, we have individualized FMF patients with symbols of different shapes. This was not possible in Figure 1 due to the number of patients, which made the figures unreadable.

We do agree with the reviewer that the genotypes are important pieces of information, we have thus decided to follow the reviewer advice and present the figures annotated for each patient genotypes in Appendix Fig S1 and S4 in addition to providing in the figure legend the genotype of the patient whenever this is feasible.

4. The authors do not describe why non-parametric tests were used. Did they check data distribution and data-transforms to then use parametric tests if possible? Even so, as they have performed non-parametric tests, they must show medians and quartile ranges and not means/SEM. Also, Some data panels lack legends e.g. black and red in Fig 3A-B.

In the first version of the manuscript, we had performed the D'Agostino-Pearson omnibus normality test and had concluded that most values did not follow a Gaussian distribution and had thus decided to perform non-parametric tests in all the data sets.

Following the reviewer comment, we have re-evaluated the data using 1) graphical methods (histogram and QQplot-see below for FMF patients). These graphical methods confirm the non-normality at least for the data presented in Fig. 1A.

Despite a tentative of log normalization of data and since data were not normally distributed, non-parametric tests were used to compare respectively quantitative and qualitative variables.

Of note, while values from FMF patients did follow a normal distribution in some figure panels (e.g. Fig.1 B UCN-01) as determined by D'Agostino-Pearson omnibus normality test, the values from HD did not. We thus used non-parametric tests that do not assume normal distribution.

The statistic section has been extensively developed to indicate our decision tree.

“For HD/patient data, normal distribution was verified using D'Agostino-Pearson omnibus normality test. When Gaussian distribution could not be demonstrated for the compared variables, non-parametric tests were used. Results were expressed as median °" interquartile range. When two groups (e.g. HD and FMF), including at least one that did not pass the normality test, were compared, Wilcoxon rank-sum test were performed, twotailed p-values are shown. When two groups (e.g. HD and FMF) that passed the normality test, were compared, unpaired t-test were performed, two-tailed p-values are shown. When multiple comparisons between groups (e.g. HD and FMF) that all passed the normality test, were made, One -way ANOVA with Sidak's multiple comparison tests were performed.

When multiple comparisons between groups (e.g. HD and FMF), some of them that did not pass the normality test, were made, Kruskal-Wallis with Dunn's multiple comparison tests were performed. Multiplicity adjusted P values are shown. When multiple comparisons between paired groups of identical numbers (e.g. FMF and FMF + inhib.), some of them that did not pass the normality test, were made, Friedman with Dunn's multiple comparison tests were performed. Multiplicity adjusted P values are shown. When a comparison between two paired groups (e.g. FMF and FMF + inhib.), at least one of each did not pass the normality test, was made, Wilcoxon matched-pairs signed rank test was used. Twotailed p value is shown. ...”

As suggested by the referee, we have modified the figure and instead of mean +/-SD, we show median and interquartile range. We apologize for the missing legends, these have been added in each panel

5. Authors should develop the molecular mechanistic side of their manuscript. 5A-For example, is Pypin dephosphorylated in patient-derived cells upon PKCinhibitor treatment (Phos-tag gels or

their phospho-antibody?) - this has not been formally demonstrated in this work.

Due to the limited amount of blood available from patients, our strategy was as follow: discover and describe a phenotype in primary human cells (Fig 1 to 3 and 6 of the revised manuscript) and perform most of the mechanistic on a genetically tractable system.

As previously mentioned (Main comment point 1, reviewer 2), the molecular mechanistic side has been strengthened in the revised manuscript.

5B-How do homozygous and/or mixed mutations affect this? Have the authors attempted to reconstitute MEFV-/- cells with other mutations or mixed mutations (e.g. express two variants with a P2A self-cleavage tag) that match their patient samples - this would be needed if they wish to draw broad conclusions that PKC inhibitors can distinguish FMF patients.

The idea of expressing two variants with a P2A self-cleavage tag is a very smart idea. Unfortunately due to the size of the *MEFV* ORF, of the pINDUCER 21 backbone and considering the lentivirus encapsidation limit, this would not be feasible/efficient in our experimental system.

As mentioned above and in the manuscript, the message of this manuscript is not to draw broad conclusions applying to each *MEFV* mutations/variants. This is explained in the discussion. In an effort to extend our data and to take into account the reviewer comment, we now present in the revised manuscript (Fig EV4) three additional *MEFV* variant (p.M680I, p. M694I and p.P369S). These results demonstrate that the response to UCN-01 can discriminate different *MEFV* variants.

6. As some of their data conflict previous data on colchicine efficacy, they should address this as they have access to patient-derived cells and good molecular cell-line systems and requisite tools.

Have they checked other inhibitors, concentrations curves, silencing of PKN1/2? There is evidence that the mouse Mefv726 mutants interact with PKN1 more after colchicine treatment. This would indicate that colchicine is not necessary acting at a second step, but increases inhibitory phosphorylation of Pyn. Mechanistically, does the M694V/S242A not interact with 14-3-3 proteins (this has been suggested before Park et al)? Does phosphorylation status/interaction with 14-3-3 of Pyn/M694V etc. change after colchicine treatment - is that the reason colchicine works in some cases and not others? Such experiments will provide more insight on this very interesting molecule.

We now present in the revised manuscript (Fig EV2AB) that nocodazole (as previously described in PBMC from HD exposed from TcdA) inhibits UCN-01-mediated inflammasome activation while Taxol does not.

We now present in the revised manuscript (Fig EV2CD) the dose response of colchicine demonstrating similar dose-response to inhibit TcdA-mediated inflammasome response in HD and UCN-01-mediated inflammasome response in FMF patients.

We now present in the revised manuscript the results of *PNKI* invalidation coupled to *PNK2* siRNA in Fig EV5. These results support our conclusions based on the use of UCN-01.

We now present in the revised manuscript a figure (Fig EV6) demonstrating that colchicine inhibits cell death of a phospho-null mutant (S242R/S208C/M694V), this result suggests that colchicine acts downstream of the dephosphorylation.

We have not addressed the role of 14-3-3 proteins that have been previously reported. Understanding why colchicine inhibits UCN-01-mediated response but not TcdA/B response is a fascinating project but we feel it is out of the scope of the current manuscript.

Referee #3 (Remarks for Author):

In the manuscript by Magnotti and colleagues a novel mechanism explaining pyrin hyperactivation in patients with FMF is proposed. The authors postulate that FMF mutant Pyrin can be activated solely by dephosphorylation at position Ser242. They demonstrate by using small molecule inhibitors of PKC superfamily kinases that FMF mutant pyrin can be activated, yet the same

inhibitors (under the conditions chosen) fail to activate wild-type pyrin. This data suggests that in FMF patients one of two safeguard mechanisms, that normally prevents pyrin activation, is missing.

The paper is of potential interest to the community as it brings new mechanistic insights into a fairly common genetic disease. Very few papers have provided a similarly deep analysis of human pyrin activation in primary patient-derived cells. While the work is of interest a number of issues require attention. Most critically a clear demonstration that the proposed mechanism is indeed correct is missing.

We thank the referee for its evaluation of the interest of our manuscript. As described in details below we have strengthened our demonstration of the proposed mechanism.

Major points:

1- The authors make the claim that pyrin dephosphorylation of S242 in FMF mutant pyrin is the mechanism by which PKC inhibitors can activate mutant pyrin. A clear biochemical characterization that these inhibitors act directly by inhibiting pyrin dephosphorylation needs to be performed. The WB shown is not very convincing. Addition of UCN-01 already reduces the amount of pyrin. What is the basis for that? It is also not clear which of the multiple bands that show up in the WB is in fact pyrin. The authors should generate a non-phosphorylatable form of pyrin and test the function and the validity of the pPyrin Ab.

First, we would like to make the point crystal clear that we can mimic PKC inhibitors action by mutating specifically the Serine 242 and Serine 208 of Pyrin (see figure 5 of the revised manuscript). We thus have the genetic evidence demonstrating that the phenotype we describe in this manuscript is specifically due to the action of the inhibitor on Pyrin. We have strengthened this point by adding in the revised manuscript the p.S208C mutant, and the S208C/S242R double mutant either in the context of WT pyrin or of M694V pyrin, and by adding the IL-18 result in addition to the cell death result.

As suggested by the referee, we have generated a S242R Pyrin-expressing cells. In the revised manuscript, we provide in Appendix Figure S7, a Western blot demonstrating the specificity of the phosphoPyrin antibody.

The Pyrin antibody is highly specific as demonstrated in Fig EV3. Indeed, the different bands on the Western blot are absent without doxycycline and are induced in a time-dependent manner upon doxycycline addition. These multiple bands have also been described by Jae Jin Chae and colleagues in PBMCs (Chae *et al*, 2008). The Western blot from their paper (Fig 2D) is shown below. This pattern is very similar to what we observed with a band around 100 kDa, discrete bands around 75kDa, a band around 50kDa and a band around 37 kDa. Dephosphorylation and the expected release from the 14-3-3 proteins is likely to accelerate the degradation of the Pyrin protein.

The reference and the following sentence have been added in the revised manuscript.

*“The Pyrin immunoblot pattern obtained upon doxycycline addition was similar to the pattern previously described in PBMCs (Chae *et al*, 2008) with a major cleavage band around 50kDa (Figure EV3B).”*

2- It is interesting that colchicine can block PKC induced FMF Pysin activation. What is the mechanism? Can the authors test whether colchicine interferes with pyrin dephosphorylation?

We now present data in Figure EV6 indicating that colchicine blocks the cell death upon doxycycline-mediated induction of the p.S208C/S242R/M694V protein (phospho-null mutants). We believe this result indicates that colchicine acts independently of Pysin dephosphorylation. The following text has been added in the result section:

“Interestingly, colchicine addition could block the cell death induced by the doxycycline-mediated expression of pS208C/M694V-, pS242R/M694V, pS208C/S242R/M694V-Pysin variants (Fig EV6), suggesting that colchicine acts independently (and downstream (Van Gorp et al, 2016; Gao et al, 2016)) of Pysin dephosphorylation.”

3- The spread of "responders" to the PKC compound for the FMF patients is very large and there are clearly some that don't respond to the treatment at all. I think this demonstrates that at least for some donors both, the proposed mechanism is incorrect as well as that the advocated test will not be as conclusive as proposed by the authors.

First, we would like to emphasize that the fast cell death is more discriminative than IL-1 β in our experience (see Fig 1 and 6 of the revised manuscript). Second, we would like to emphasize that our cohort of patients is heterogeneous and includes some patients with a mono-allelic *MEFV* mutation. In an effort to better delineate the specificity of the test/response to PKC superfamily inhibitors, we now present in Appendix Figure S1, the genotype of each patient in the different panels of figure 1.

As explained in the response to reviewer 2, we have been cautious explaining that the test should be further explored before validation. FMF is highly heterogeneous both in terms of *MEFV* variants and copy number of pathogenic variants. Future studies are required to delineate the potential use of the test. We believe this is clear in the discussion of the revised manuscript.

As a further validation of the proposed mechanism and of its potential limits, we have further tested two clearly pathogenic *MEFV* variants: p.M694I, p.M680I and a *MEFV* variant of uncertain significance p.P369S in our U937 cell model. The results now presented in Fig EV4 demonstrate that the two cell lines expressing the two pathogenic variants activate the Pysin inflammasome in response to UCN-01 while the cells expressing p.P369S behave as WT Pysin-expressing cells.

4- PKC inhibitors can promote noncanonical caspase-8 dependent release of IL-1 β . Can the authors exclude the reported inflammasome independent non-canonical processing of IL-1 β ?

As suggested by the referee (minor point below), we have tested caspase-1, -3 and -8 inhibitors in primary human monocytes. The results presented in the revised manuscript in Fig EV1G strongly suggest that the response is caspase-1-mediated and not caspase-8 mediated.

Furthermore, based on the genetic invalidation of *CASP1* in U937 cells expressing p.M694V cells (Figure 4E), we strongly favour a major role of caspase-1 in IL-1 β processing and release.

Minor points:

1- The data representation is inconsistent. Please select one representation style. It would be good if for the inhibitor experiments would make use of unique symbol or colors for each of the donors to know which comparison to make.

We apologize for the inconsistency in the representation in the previous version. This has been corrected in the revised manuscript.

As suggested by reviewer 2 and 3, one unique symbol for each patient has been used in Figure 2B-C, 3A-E of the revised manuscript to allow for comparison. Due to the number of patients in each panel in Figure 1, we have decided to present the data with each patient and its genotype individualized in Appendix Fig S1.

2- Genetic evidence for the proposed mechanism is missing. The authors report that the attempts to knock-out *PKN1/2* failed. Yet, there are alternative ways to generate a transient knock down. This

work is essential, especially as the inhibitors used have many effects.

First, we have to disagree with the reviewer since we used in the previous version a phosphonull mutation (S242R) combined to the p.M694V mutation to provide genetic evidence of the proposed mechanism. This point has been strengthened in the revised manuscript (Fig 5 of the revised manuscript) by mutating the other phosphorylated serine (S208) and by creating the double phospho-null mutant (S208C/S242R) each one either on a WT *MEFV* background and on a p.M694V *MEFV* background.

Working with kinases is challenging due to redundancy (*PKN1/2* and potentially *RSK2,1,3*) and lethality associated with *PKN1/PKN2* invalidation. We present data in Fig EV5 of the revised manuscript with knock-out of *PKN1* and knock-down of *PKN2*. Although the levels of cell death obtained were not very high (certainly not reaching the levels of the triple p.S208C/S242R/M694V), they were specific of i) doxycycline-mediated induction of Pyn ii) Induction of p.M694V Pyn iii) efficient siRNA against *PKN2*. We thus feel these data support our proposed mechanism.

Specific for figures:

1-Figure 2:

- Please include all staining controls for the ASC specks? This could at least be added to the supp figures. A quantitation of speck like objects in the secondary alone control cells would be appreciated.

The staining controls have been added in Appendix Fig S2. Quantitation of speck-like objects in control cells stained with both primary and secondary antibodies shows less than 5% of such objects (see Fig 2B first two bars).

2- Inhibition using VX-765 and ZVAD would be useful to further define the type of cell death and back up their current data.

We now present in the revised manuscript Fig EV1G, use of inhibitors to define the caspase involved in IL-1 β release. Unfortunately, caspase inhibitors are largely inefficient to block cell death (see for example in (Broz *et al*, 2010) " inhibitor treatment only modestly reduced host cell death in WT macrophages (Figure3B). This was likely due to incomplete inhibition by Z-YVAD-FMK...) and we did not observe substantial cell death reduction upon using YVAD-FMK or VX-765.

Furthermore, the role of caspase-8 in cell death is challenging to test since caspase-8 inhibition triggers necroptosis (which we have observed in primary monocytes treated with IETD-FMK) and requires to work of a RIP3KO background in order to invalidate caspase-8 (Kaiser *et al*, 2011).

The visualisation of the ASC specks (Fig 2A-B), the detection of active caspase-1 (Fig 2C), the lack of Annexin-V+PI- cells (Fig 2D), combined to the result of the genetic invalidation of *CASP1* and *GSDMD* in U937 cells expressing p.M694V (Fig 4C) strongly suggest that cell death is caspase-1-mediated pyroptosis.

Figure 3:

3- Controls are needed for the effect of colchicine.

The effectiveness of colchicine is demonstrated by the full inhibition of IL-1 β in HD monocytes exposed to TcdB.

The effect of colchicine has been strengthened in the revised manuscript by presenting the dose response (Fig EV2C-D) and another inhibitor (nocodazole) (Fig EV2A-B).

4- Vector alone control is lacking for all experiments

The vector alone control has been added in the revised Fig. 4A, B, D. It is fully in line with the "No Dox" control.

5- Figure 4D, how do the M694V + Dox cells secrete Il-1 in the absence of an LPS priming signal (and quite a lot, 500 pg/ml)?

This is likely due to the use of PMA for U937 cells differentiation. We do see higher IL-1 β in M694V-expressing cells than in WT Pyrin-expressing cells in presence of PMA or PMA+ LPS. This is reminiscent of what has been described on primary FMF cells by Omenetti and colleagues (Omenetti *et al*, 2014) but we have decided not to focus on this phenotype in this study.

The corresponding sentence has been modified in the revised manuscript and reads as presented below:

“the differences between WT and p.M694V Pyrin-expressing cell lines in terms of IL-1 β secretion were largely specific for PKC inhibitors, although low levels of IL-1 β and IL-18 secretion were observed in p.M694V Pyrin-expressing cells in absence of PKC inhibitors, possibly due to the use of PMA”

6- Figure 4G - Do the authors change the scale of the graph for the UCN-01 addition? At this point eh p.S242R seems to change position relative to the WT (shows lower values rather than higher).

This refers to figure 5A of the revised manuscript, which is a new figure.

For your information, the scale of the graph was not changed. This is an artefact due to the arrest of the program and the restart (with a shaking step). We have decided to correct for this artefact by correcting the values after UCN-01 addition so that values right before and right after UCN-01 addition were identical. The correction has been made clear both in the figure legend and in the material and method section.

7- Figure 4 H would be clearer if they labeled the legend fully. Furthermore, Wilcoxon rank-sum test (Mann-Whitney U test) requires that the samples are non-related. For related or matched samples (in a not normally-distributed population), a Wilcoxon signed-rank test is more convenient. This is the case when comparing Dox and Dox+UNC-01 conditions in the same sample.

Full legends have been added in all the figures.

The reviewer is right regarding pairing of the data. Due to multiple comparisons, we have used Friedman test with Dunn's corrections (a paired, non-parametric test).

References

- Broz P, von Moltke J, Jones JW, Vance RE & Monack DM (2010) Differential requirement for Caspase-1 autoproteolysis in pathogen-induced cell death and cytokine processing. *Cell Host Microbe* **8**: 471–83
- Chae JJ, Wood G, Richard K, Jaffe H, Colburn NT, Masters SL, Gumucio DL, Shoham NG & Kastner DL (2008) The familial Mediterranean fever protein, pyrin, is cleaved by caspase-1 and activates NF-kappaB through its N-terminal fragment. *Blood* **112**: 1794–1803
- Gaidt MM, Ebert TS, Chauhan D, Schmidt T, Schmid-Burgk JL, Rapino F, Robertson AAB, Cooper MA, Graf T & Hornung V (2016) Human Monocytes Engage an Alternative Inflammasome Pathway. *Immunity* **44**: 833–846
- Gao W, Yang J, Liu W, Wang Y & Shao F (2016) Site-specific phosphorylation and microtubule dynamics control Pyrin inflammasome activation. *Proc. Natl. Acad. Sci. U. S. A.* **113**: E4857–4866
- Jamilloux Y, Lefeuvre L, Magnotti F, Martin A, Benezech S, Allatif O, Penel-Page M, Hentgen V, Seve P, Gerfaud-Valentin M, Duquesne A, Desjonqueres M, Laurent A, Remy-Piccolo V, Cimaz R, Cantarini L, Bourdonnay E, Walzer T, Py BF, Belot A, et al (2017) Familial Mediterranean fever mutations are hypermorphic mutations that specifically decrease the activation threshold of the Pyrin inflammasome. *Rheumatol. Oxf. Engl.*
- Kaiser WJ, Upton JW, Long AB, Livingston-Rosanoff D, Daley-Bauer LP, Hakem R, Caspary T & Mocarski ES (2011) RIP3 mediates the embryonic lethality of caspase-8-deficient mice. *Nature* **471**: 368–72
- Man SM, Hopkins LJ, Nugent E, Cox S, Gluck IM, Toulomousis P, Wright JA, Cicuta P, Monie TP & Bryant CE (2014) Inflammasome activation causes dual recruitment of NLRC4 and NLRP3 to the same macromolecular complex. *Proc. Natl. Acad. Sci. U. S. A.* Available at: <http://www.ncbi.nlm.nih.gov/pubmed/24803432>
- Omenetti A, Carta S, Delfino L, Martini A, Gattorno M & Rubartelli A (2014) Increased NLRP3-dependent interleukin 1beta secretion in patients with familial Mediterranean fever: correlation with

MEFV genotype. *Ann. Rheum. Dis.* **73**: 462–469

Park YH, Wood G, Kastner DL & Chae JJ (2016) Pyrin inflammasome activation and RhoA signaling in the autoinflammatory diseases FMF and HIDS. *Nat. Immunol.*

Pierini R, Juruj C, Perret M, Jones CL, Mangeot P, Weiss DS & Henry T (2012) AIM2/ASC triggers caspase-8-dependent apoptosis in Francisella-infected caspase-1-deficient macrophages. *Cell Death Differ.* **19**: 1709–21

Sagulenko V, Thygesen SJ, Sester DP, Idris A, Cridland JA, Vajjhala PR, Roberts TL, Schroder K, Vince JE, Hill JM, Silke J & Stacey KJ (2013) AIM2 and NLRP3 inflammasomes activate both apoptotic and pyroptotic death pathways via ASC. *Cell Death Differ.* **20**: 1149–1160

Van Gorp H, Saavedra PHV, de Vasconcelos NM, Van Opdenbosch N, Vande Walle L, Matusiak M, Prencipe G, Insalaco A, Van Hauwermeiren F, Demon D, Bogaert DJ, Dullaers M, De Baere E, Hochepeid T, Dehoorne J, Vermaelen KY, Haerynck F, De Benedetti F & Lamkanfi M (2016) Familial Mediterranean fever mutations lift the obligatory requirement for microtubules in Pyrin inflammasome activation. *Proc. Natl. Acad. Sci. U. S. A.*

2nd Editorial Decision

22 August 2019

Thank you for the submission of your revised manuscript to EMBO Molecular Medicine, and please accept my apologies for the delay in getting back to you, which is due to the fact that one referee needed more time to complete his/her report. We have now received the three reports, which are supportive of publication. I am therefore pleased to inform you that we will be able to accept your manuscript pending minor editorial amendments as well a response to referee #2.

At this stage, we'd like you to discuss the referee's points in writing or to make changes in the figures. We do not ask you to provide any additional experiments at this stage.

I look forward to reading a new revised version of your manuscript as soon as possible.

***** Reviewer's comments *****

Referee #1 (Remarks for Author):

The authors have satisfactorily addressed our concerns and remarks.

Referee #2 (Comments on Novelty/Model System for Author):

The cell line model and revised statistics satisfactorily address concerns raised in review version 1.

Referee #2 (Remarks for Author):

In their revised version, Magnotti et al have addressed many of the main points and I am largely satisfied. However, I have the following points on new data which should be addressed.

1. The IP/IB (Fig. S7) showing loss of phosphorylation of mutant PYRIN is not convincing. I could not see bands corresponding to S242R and S242R/M694V in the IP blot on right. There is a ~100kDa band in first two lanes and not lanes 3 and 4. Therefore it is not surprising to see no band in blot on left - am I missing something here?

2. Data in Fig 5. could be separated into graphs that only show the S208C and S242R. There are too many PI uptake curves to be able to read the realtime assay data meaningfully.

3. More broadly, figures (including EV and Supplementary) are not prepared to a good standard and contain different serif and non-serif fonts, poor choice of colours and too small shapes. I urge the authors to stick to one clear and simple style throughout. For example, X and Y axes labels in Fig 1 are serif or non-serif., 1D and 1G have uneven axes, patient genotype in 2A could also be on the figure panel to make it easier, 2E pink/purple are difficult to distinguish, and so on.

4. Is it possible that DNA-bound PI degrades in dead/lysed cells which therefore lose fluorescence over time? Shouldn't assays with Nigericin, for example in 4B, be stopped at 1-2h anyway and shown as AUC bar graphs which are easier to read with so many conditions?

Minor comment:

1. pg6 check reference format for Tamaoki et al.
2. pg 7 where they say 'normalized', I think the authors mean "which was similar at higher concentrations"?
3. I feel that data from EV6 should be shown in Fig 5 as area under the curve bar graphs for simplicity as this nicely makes the authors' point on spontaneous cell death.
4. pg 16 "IL-1b dosage" should be changed to "IL-1b release"?
5. I think it would be good to add equally spaced minor/major ticks to Y-axis in Fig 1D and 1G and related elsewhere.

Referee #3 (Remarks for Author):

All my concerns have been addressed.

2nd Revision - authors' response

4 September 2019

***** Reviewer's comments *****

Referee #1 (Remarks for Author):

The authors have satisfactorily addressed our concerns and remarks.

Referee #2 (Comments on Novelty/Model System for Author):

The cell line model and revised statistics satisfactorily address concerns raised in review version 1.

Referee #2 (Remarks for Author):

In their revised version, Magnotti et al have addressed many of the main points and I am largely satisfied.

However, I have the following points on new data which should be addressed.

1. The IP/IB (Fig. S7) showing loss of phosphorylation of mutant PYRIN is not convincing. I could not see bands corresponding to S242R and S242R/M694V in the IP blot on right. There is a ~100kDa band in first two lanes and not lanes 3 and 4. Therefore it is not surprising to see no band in blot on left - am I missing something here?

As described in the main manuscript and in the previous rebuttal letter, Pyrin is susceptible to processing/degradation. UCN-01-mediated dephosphorylation and S208C/S242R phosphonull mutations increase this processing/degradation likely due to the loss of interaction with the phospho-dependent chaperone proteins from the 14-3-3 family.

While it is thus difficult to compare the phosphorylation of WT and mutated S242R full length Pyrin, the phosphorylation of Pyrin is still obvious in the 50kDa degradation/processed fragment allowing to validate the specificity of the Phospho-S242 antibody. The following sentence is present in the legend of the Appendix Fig S8:

“Note that the two degradation products around 50 kDa are heavily phosphorylated on S242 in both WT and p.M694V-expressing cells while the phosphorylation is undetectable in p.S242R Pyrin-expressing cells.”

2. Data in Fig 5. could be separated into graphs that only show the S208C and S242R. There are

too many PI uptake curves to be able to read the realtime assay data meaningfully.

We have split the graph 5A in two graphs. The following sentence has been added in the legend of the revised Fig.5.

“Data presented in A and B are from the same experiment and have been split for clarity. The WT and p.M694V controls are duplicated in A and B.”

3. More broadly, figures (including EV and Supplementary) are not prepared to a good standard and contain different serif and non/serif fonts, poor choice of colours and too small shapes. I urge the authors to stick to one clear and simple style throughout. For example, X and Y axes labels in Fig 1 are serif or non-serif., 1D and 1G have uneven axes, patient genotype in 2A could also be on the figure panel to make it easier, 2E pink/purple are difficult to distinguish, and so on.

We have made the following modifications:

- Police was changed to Helvetica.
- In Fig 1D and 1G the number 20% (which was there to indicate how was calculated the time to reach 20% cell death) was removed.
- Patient genotypes were added in Figs 2A and 2D
- The color has been changed in Fig 2E
- Fig 5A has been split in two panels.
- The origin of onsets has been added in Fig 2A and the size of the arrowheads has been increased.
- The shape of DC patients'symbol has been corrected in Fig 6D
- Police size has been increased in Appendix Fig S1 and S5

The size of the shapes could not be increased without affecting the clarity of the graphs.

4. Is it possible that DNA-bound PI degrades in dead/lysed cells which therefore lose fluorescence over time? Shouldn't assays with Nigericin, for example in 4B, be stopped at 1-2h anyway and shown as AUC bar graphs which are easier to read with so many conditions?

DNA-bound PI might indeed degrade which could explain the loss of fluorescence over time. We have decided to keep the same scale for all the cell death assays presented in Fig. 4A, B, C. AUC versus kinetics is a recurrent discussion with colleagues and reviewers. AUC are easier to read but are associated with a loss of information (two very different shapes can result in an identical AUC). In Figure 4, there are up to 5 curves and we feel the panels are easy to read and have thus decided not to present the AUC. We have added the AUC bar graph in Fig EV5B as suggested by the reviewer (see minor point 3 below).

Minor comment:

1. pg6 check reference format for Tamaoki et al.

Done

2. pg 7 where they say 'normalized', I think the authors mean "which was similar at higher concentrations"?

We have modified the manuscript as indicated below:

“The hyper-responsiveness of FMF monocytes to PKC superfamily inhibitors thus differs from their hyper-responsiveness to Clostridioides difficile toxin TcdB, which was observed only at low doses of TcdB (Jamilloux et al, 2017).”

3. I feel that data from EV6 should be shown in Fig 5 as area under the curve bar graphs for simplicity as this nicely makes the authors' point on spontaneous cell death.

We have kept the colchicine data in Expanded View Fig5 since the current Fig5 is already complex. As suggested by the reviewer, we have added the AUC bar graph in Fig EV5B.

4. pg 16 "*IL-1b dosage*" should be changed to "*IL-1b release*"?

Done

5. *I think it would be good to add equally spaced minor/major ticks to Y-axis in Fig 1D and 1G and related elsewhere.*

We have removed the 20% value to keep equally spaced ticks in Fig. 1D, 1G, 6C.

Referee #3 (Remarks for Author):

All my concerns have been addressed.

Corresponding Author Name: Thomas Henry
 Journal Submitted to: EMBO Molecular Medicine
 Manuscript Number: EMM-2019-10547